# A Review of Recent Advances in Human-Motion Energy Harvesting Nanogenerators, Self-Powering Smart Sensors and Self-Charging Electronics

**DOI:** 10.3390/s24041069

**Published:** 2024-02-06

**Authors:** Justyna Gołąbek, Michał Strankowski

**Affiliations:** Department of Polymer Technology, Faculty of Chemistry, Gdańsk University of Technology, 80-233 Gdańsk, Poland; justyna.golabek@pg.edu.pl

**Keywords:** human-motion energy harvesting, triboelectric/piezoelectric nanogenerator, wearable electronics, implantable devices

## Abstract

In recent years, portable and wearable personal electronic devices have rapidly developed with increasing mass production and rising energy consumption, creating an energy crisis. Using batteries and supercapacitors with limited lifespans and environmental hazards drives the need to find new, environmentally friendly, and renewable sources. One idea is to harness the energy of human motion and convert it into electrical energy using energy harvesting devices—piezoelectric nanogenerators (PENGs), triboelectric nanogenerators (TENGs) and hybrids. They are characterized by a wide variety of features, such as lightness, flexibility, low cost, richness of materials, and many more. These devices offer the opportunity to use new technologies such as IoT, AI or HMI and create smart self-powered sensors, actuators, and self-powered implantable/wearable devices. This review focuses on recent examples of PENGs, TENGs and hybrid devices for wearable and implantable self-powered systems. The basic mechanisms of operation, micro/nano-scale material selection and manufacturing processes of selected examples are discussed. Current challenges and the outlook for the future of the nanogenerators are also discussed.

## 1. Introduction

In today’s modern society, the smartphone is the world’s most popular electronic device, and has become an integral part of our daily lives. According to IDC’s Worldwide Quarterly Mobile Phone Tracker, by 2025 the total number of mobile phones sold worldwide will reach up to 1708.0 million units, resulting in an annual growth rate of 1.2% compared to 2020 [1]. However, while their production grew, even more popular technological innovation trends appeared—personal wearables and Internet of Things technology. Smartwatches, smart glasses, fitness trackers, smart clothes and so on are gaining increasing attention due to their wide range of applications [2]. According to *The Global Industrial Wearable Device Market Research Report*, the Global Market is estimated to reach more than USD 2329 million by 2027 versus USD 1373 million in 2020 [3]. Portable devices such as smartphones, tablets, laptops and wearables have one specific feature—they need a power source such as a battery or supercapacitor to function correctly. The battery is often rechargeable and can be mounted inside the device or removed. It accumulates and then stores the energy required to power an electronic device during charging. Since human personal devices are usually small and lightweight, battery size and capacity are also limited. As a result, a battery with a finite lifetime needs to be constantly recharged, which may not always be possible. Moreover, some of their components are hazardous chemicals and non-biodegradable materials; thus, environmental unfriendliness is another drawback of traditional batteries [4,5]. To solve the problems mentioned above, many researchers have started looking for renewable and sustainable energy sources that would enable the continuous operation of electronics and further contribute to overcoming the energy deficiency crisis [6,7]. As a result, energy harvesting technology that harnesses energy from the environment or external sources and converts it into electricity has been developed [8]—radio frequency [9], thermal energy [10], tidal energy [11], solar energy and wind energy [12], and kinetic energy related to human body movements or thermal energy [13].

One of the main disadvantages of atmospheric sources (tidal, solar, wind) is their variable nature [14]. Depending on the weather conditions and occurrence in different geographical regions, they cannot always be effectively utilized and generate enough energy to power personal portable and wearable devices [15]. However, the human body as an energy source generates a considerable amount of natural energy, which is often wasted [16]. In 1996, Starner calculated that a 68 kg adult male could generate up to 100 W of power during walking, moving his arms and fingers, and breathing, as shown in Figure 1 [17].

Theoretically, once the energy from these simple, effortless tasks is harvested, it should be enough to power small electronics, for example, smartphones, music players or digital cameras, as their power requirements are in the milliwatt to a few watts range. However, many practical aspects are very problematic to solve. The main issue related to energy extraction from the human body is the device’s design, which must be flexible, wearable, and operate following body movements [18]. As a solution to this issue, many types of generators have been introduced and categorized according to the energy source, as listed in Table 1 [19].

Chemical reactions and electron transfer in chemical substances typically follow chemical-based human-body-energy harvesting. In our opinion, a great example of such technology is a biofuel cell (BFC), which uses human body fluids—glucose and lactase acid—as energy sources, and converts electrical energy through their redox reaction. Depending on the catalysts used, there are two types of BFC—enzymatic fuel cells (EFCs), which work based on oxidation and reduction reactions and microbial fuel cells (MFCs), which use living microorganisms [20]. For example, they have been used in the medical field, as the power supply for biosensors [21,22] or a power source for wireless sensor networks [23].

One of the most significant advantages of chemical energy used primarily in physical sensors is that they provide physiological parameters that cannot be provided with other techniques and are highly sensitive to biomarkers. However, many disadvantages need to be discussed. First of all, without encapsulation, cross-contamination by environmental pollutants is possible; they are sensitive to temperature and humidity fluctuations and are challenging to fabricate and integrate as a wearable device. Second, in many cases, they need to be externally powered, the efficiency is low, and the power generated is unreliable and inconsistent [24]. Even though the concept of biofuel cells (nowadays interchangeably named biobatteries) has been studied for many years, and there are some interesting articles about them [25,26,27], from our point of view, the actual application perspective and commercial use are still not within range. Thus, chemical energy sources will not be the subject of discussion in this review.

Thermal-based energy harvesting techniques such as thermoelectric and pyroelectric generators are based on temperature differences between two surfaces. Thermoelectric modules can be power generators based on the Peltier effect (cooler) or the Seebeck effect [28]. A thermoelectric energy harvester, which works according to the Seebeck effect, uses temperature gradients between two materials—one of them a p-type semiconductor with surplus holes and the second one an n-type semiconductor with surplus electrons—to cause an electron flow. When a temperature gradient grows higher between two surfaces (one hot and one cold) through thermoelectric materials, a free-charge movement converts thermal energy into electrical energy [29]. Thermoelectric generator output performance strongly depends on the materials’ characteristics and their appropriate selection. The most commonly used thermoelectric materials are bismuth telluride (Bi_2_Te_3_), antimony telluride (Sb_2_Te_3_), zinc antimonite (ZnSb), and silicon germanium (SiGe). Recently, novel materials such as conjugated polymers PEDOT:PSS, polymers consisting of metal ions and ligands, carbon nanotubes and graphene have been introduced to enhance the output performance of generators [24]. The advantages of thermoelectric generators include low cost and maintenance, high reliability and longevity, scalability, and accessibility. As for disadvantages, low power density and energy conversion efficiency, the unstable character of harvested power and the need for a boost converter should be mentioned [30]. In comparison, the pyroelectric effect is based on the spontaneous re-orientation of electric dipoles of polar materials which undergo time-dependent temperature fluctuations (dT/dt), resulting in an alternating current [31]. Pyroelectric materials temporarily change their spontaneous polarization (P_S_) with the temperature changes in the materials. So, when the spontaneous polarization decreases due to thermal vibration, the surface charge decreases when the temperature increases. Electrical current flow occurs in the external circuit when pyroelectric materials are under short-circuit conditions [31]. Triglycine sulfate (TGS), PVDF, lithium tantalate (LiTaO_3_), zinc oxide (ZnO) and lead-free perovskite ceramics (for example, bismuth sodium titanate-barium titanate, BNT-BT) are leading representatives among the pyroelectric materials [32]. The advantages are a wide range of operating temperatures and small form, and the main disadvantage is the limitation of the pyroelectric generators caused by the frequency in temperature fluctuation. As for the application prospects, thermoelectric and pyroelectric generators found application as energy harvesting devices, powering low-power electronics in healthcare and therapeutic fields, especially as sensors. Precise descriptions of working mechanisms, materials and applications for both generators can be found in the following articles and reviews [28,32,33,34,35,36,37,38,39].

However, considering their relatively low power output, the kinetic energy generated by humans is more interesting and prosperous as a potential power source for wearable and implantable devices. Therefore, after detailed research and considering future trends, we decided that this review will focus most of all on the biomechanical-based energy sources and techniques based on the piezoelectric effect [18,40] and triboelectric effect [41], which includes piezoelectric nanogenerators (PENGs) [42,43] and triboelectric nanogenerators (TENGs) [44,45]. Although they have been extensively researched over the past few years and significant progress has been made, some drawbacks still prevent further development. Namely, PENGs require fabrication over a large area, and TENGs can be subject to mechanical damage [46]. Moreover, since TENGs have a low power density, continuously generating sufficient energy and directly powering electronic devices without first storing the converted energy is impossible without the help of supercapacitors and batteries [47]. However, depending on the ultimate purpose, new ideas have been developed that show promising performance, including hybrid “all in one” systems that are a combination of two or more different harvesters, fibre-based harvesters [48] or a modification of already existing designs using new polymers and nanocomposites displaying self-healing or shape-memory capabilities [48].

This paper reviews recent reports of self-powered piezoelectric and triboelectric nanogenerators for energy harvesting. First, the working principles and operational modes are discussed, along with the designs and configurations of the energy harvesting device systems. The range of materials for both energy harvesting systems is briefly described. Then, attention is paid to specific applications towards wearable and implantable self-powered devices, especially in the medical field. Then, the two nanogenerators are compared in terms of their advantages, disadvantages, and application possibilities. Finally, this technology’s future outlook and perspectives are briefly discussed, considering current challenges and limitations.

## 2. Working Principles and Materials of Energy Harvesting Systems

Mechanical energy, primarily obtained from human body movements, is one of the most common and abundant sources of energy, the most significant advantage of which is its independence from time and place compared to other sources of environmental energy. It can be captured, transformed into electricity, and then stored for future purposes. Conversion is possible through transduction mechanisms such as piezoelectric and triboelectric nanogenerators [49,50]. The operating principles, materials, advantages and disadvantages of each mechanism are highlighted and briefly described in the following paragraphs.

### 2.1. PENG-Based Energy Harvesting System

#### 2.1.1. PENG—Working Mechanism and Materials

The piezoelectric energy nanogenerator (PENG) is based on the piezoelectric (direct) effect associated with the induction of electric charge, which accumulates in certain piezoelectric materials as a result of applied mechanical deformation, which leads to the polarization of the electric dipole motion [51]. The direct effect determines the ability of a piezoelectric material to convert mechanical stress or strain into electrical energy. This process is reversible (indirect)—the generation of mechanical stress is possible when an electric field is applied to the materials [16]. The direct effect is used in piezoelectric sensors, while the indirect effect is used in piezoelectric actuators [52].

PENG systems are classified according to the materials used or operating modes—d_31_ and d_33_. In the d_31_ (longitudinal) operating mode, the stress/strain is applied in the axial direction, as opposed to the voltage, which is obtained in the perpendicular direction; in the d_33_ (transverse) operating mode, the applied stress/strain, as well as the generated voltage, have the same direction [18,52]. The d_33_ mode has a higher voltage output, while the d_31_ mode provides a higher output current [52]. The d_33_-mode piezoelectric energy harvester is a structure consisting of thin layers stacked on top of each other with parallel electrical and mechanical connections. The total displacement of the entire stack results from all the films in the system multiplied by the movement of each thin film. The d_31_-based piezoelectric energy harvester consists of a piezoelectric layer sandwiched between electrode layers. When an electric field is applied in the vertical direction, mechanical stress/strain will be generated in the horizontal direction. Figure 2 shows both the transverse mode (a) and longitudinal mode (b) [53].

Piezoelectric materials have a significant influence on the performance of the piezoelectric energy harvesting device. Depending on their crystal structure, they are divided into single crystals, ceramics, polymers and (nano)composites [54].

Single crystals exhibit unique piezoelectric properties due to their structure’s regular order of positive and negative ions, except for crystalline defects. Examples of the most commonly used piezoelectric materials that are used explicitly in sensors and actuators include lithium niobate (LiNbO_3_), a solid solution of lead magnesium niobate-lead titanate (PMN-PT), as well as lead zinc niobate-lead titanate (PZN-PT) [54]. However, due to the toxic nature of lead, lead-free piezoelectric ceramics are receiving more attention [55].

Piezoelectric ceramics (piezoceramics) are polycrystalline materials composed of randomly arranged small-sized crystals that exhibit piezoelectricity only after the polarization process [51]. The leading representative, characterized by a high piezoelectric effect and low dielectric loss, is often modified or doped lead-zirconate-titanate (PZT). Non-toxic piezoceramics with a perovskite crystal structure include barium titanate (BaTiO_3_), potassium niobate (KNbO_3_) and bismuth sodium titanate (BNT-BKT) [56]. Even though piezoceramics have some of the highest values for essential properties such as electromechanical coupling coefficient (k), piezoelectric strain constant (d) and dielectric constant (ε), their brittleness and stiffness pose obstacles in the field of flexible electronics. However, to achieve better stretchability and flexibility, these materials have been formed into thin films and wires, nano-sized particles, and fibres [57,58].

Carbon-based piezoelectric polymers (piezopolymers) are flexible materials that, when stretched or drawn in a uniaxial or biaxial direction, induce the transformation of a non-polar phase into a polar phase or induce polarization to the piezoelectric form [58]. They are lightweight and resistant to heavy loads, their production cost is relatively low, and many are biodegradable and biocompatible. Moreover, they generate sufficient voltage with appropriate output power [50,56]. However, the energy transformation and electromechanical coefficients are low [19]. The most intensively researched material among piezoelectric polymers is semi-crystalline polyvinylidene difluoride (PVDF) and its copolymers, such as polyvinylidene fluoride-trifluoroethylene (PVDF-TrFE). Due to its flexibility, it has been tested as a piezoelectric energy harvester for portable devices [58]. Other examples include polylactic acid (PLA) or cellulose and its derivatives [56].

Some characteristic properties of piezoelectric materials in piezoelectric composites and nanocomposites were superior to single crystals, piezoceramics and piezopolymers alone. Therefore, combining these materials and reinforcement with nanoscale fillers was investigated to improve performance and create a system with excellent piezoelectric properties typical of piezoceramics and excellent elasticity typical of piezopolymers. An excellent example of such a structure is lead-zirconate-titanate ceramic in the form of fibres, rods or particles embedded in a polymer matrix [54,58].

#### 2.1.2. PENG Energy Harvesting Device System

A piezoelectric energy harvesting device consists of two main parts—a mechanical system that produces electricity, and an electrical system with an electrical circuit that converts and rectifies the generated alternating voltage (AC) into direct voltage (DC). The efficiency of energy production depends on the device design—a properly selected piezoelectric transducer and appropriate integration with the electrical circuit [59].

There are four types of piezoelectric transducers:Unimorph or bimorph cantilever beam—this consists of one (unimorphic) active piezoelectric material layer placed between two electrodes or two (bimorphic) identical piezoelectric layers and usually a metallic layer in the form of a conductive non-piezoelectric layer placed at one end to create a model working in the bending mode [54,60]. In practice, bimorphic geometry is more commonly used because it can double the energy generated in output power without drastically modifying the device’s volume [54].Cymbal transducer—consists of a disk-shaped (usually ceramic) piezoelectric layer installed between two cymbal-shaped metal caps on either side. While the cymbal transducer is subjected to an external lateral force, the metal end caps with cavities act as a mechanical transformer, converting some of the axial stress into the radial stress, which enhances both piezoelectric coefficients—d_31_ and d_33_. Thus, the cymbal design can generate higher output energy than a cantilever-type device (up to 100 µW) [50,61].Circular diaphragm—consists of a thin disk-shaped piezoelectric layer connected to a metal shim and then clamped to the edges of a clamping ring [53]. To increase the efficiency of the piezoelectric energy harvester at lower frequency levels and increase its output power, a proof mass is attached to the centre of the diaphragm to provide tension to the piezoelectric material [58]. The THUNDER invention represents another method. THUNDER (thin-layer composite ferroelectric unimorphs driver) is a combination of active PZT layers with various inactive metallic materials with high durability and resistance to deformation [62,63].Stack configuration—consists of multiple piezoelectric layers stacked on top of each other so that the direction of polarization of each layer is consistent with the applied pressure. The stack configuration is based on the d_33_ mode, which is particularly useful in applications requiring high pressure. However, the layers require coupling with mechanical amplifiers due to the reduced mechanical energy caused by the greater stiffness of the structural configuration [54].

Selected advantages and disadvantages of each configuration are presented in Table 2.

### 2.2. TENG-Based Energy Harvesting System

#### 2.2.1. TENG—Working Mechanism and Materials

The triboelectric energy harvesting system consists of three modules—an energy harvesting module, an energy management module, and a storage module. The energy harvesting module is based on the triboelectric nanogenerator (TENG), first described by Zhong Lin Wang’s research group in 2012 [64]. The working mechanism of TENG is based on the coupling effect of triboelectrification and electrostatic induction. Triboelectrification results from frictional contact between two different materials with distinct electron affinities (metals, ceramics, polymers). Although it can be dangerous, TENG technology is used as a basis for collecting the mechanical energy in the environment and then converting it into electrical energy [65].

In general, a triboelectric nanogenerator consists of conducting electrodes installed on the back surface of two materials, with one of the triboelectric layers being able to donate electrons and the other layer being able to accept them. When these two layers are subjected to an external mechanical force and physical contact occurs, electrons are transferred, creating an equal number of opposite charges on both surfaces [66]. After removing the pressure and separating the materials, electrostatic induction induces the previously generated charges, which causes a potential difference with electrons flowing through the external circuit. Electron transfer occurs until the materials are completely separated, which leads to an equilibrium state [57]. Subsequently, repeated contact between layers reduces the potential difference, electrons in the external circuit flow in the opposite direction to maintain electrostatic equilibrium, and these continuous contact–separation cycles cause the conversion of the mechanical input signal into an alternative current signal [66].

TENGs have been extensively researched over the past ten years, resulting in the design of four types of working modes [67]:

The *vertical contact–separation* mode is the most common form of TENG. There are always two vertically arranged dielectric materials (with opposing polarities) with electrodes covering their backs. The continuous alternating contact–separation between the two layers leads to a drop in potential and electron flow through the external circuit to balance the potential difference [68]. TENGs based on this mode have a simple structure, high strength, and high-power density, and are easy to model and analyse [69]. A classic example of this mode is the sole of a shoe in wearable devices [70]. The operating mechanism of the vertical contact–separation mode based on the work of Yang et al. [71] is shown in Figure 3.

*Lateral-sliding mode*—there is no gap between the two planar layers, which, under the influence of an external horizontal force, perform sliding or rotational movements with a variable effective contact area [69]. Triboelectric charges are formed on each dielectric surface, creating a potential difference between the electrodes. The potential difference constantly depends on the changing effective contact area, generating an alternating current [72]. Compared with the previous mode, this mode creates triboelectric charges more effectively, increases output power, and no air gap is needed to separate the friction surfaces. The main disadvantage of the lateral-sliding mode is high-frequency friction, which can lead to severe material wear, reducing the service life and durability of the triboelectric nanogenerator [69]. The working mechanism of the lateral-sliding mode based on the work of Wang et al. [73] is presented in Figure 4.

*Single-electrode mode*—there is only one reference electrode (material which can easily lose electrons) connected to the ground and acting as a triboelectric layer in contact with the freely moving dielectric material (high electron affinity) [68,69]. Free electrons flow from the electrode to the ground as the charged dielectric material moves toward the electrode. As the charged dielectric material moves away from the electrode, free electrons flow back from the ground towards the electrode. These periodic contact–separation movements between the dielectric material and the grounded electrode produce alternating current [69]. This design is used, for example, in a snowfall energy collector [74], strain sensors [75] and angle measurement sensors [76]. The main drawback of the single-electrode mode is the low electron-transfer efficiency, which affects the final output power obtained [69]. The working mechanism of the single-electrode mode based on the work of Wang et al. [77] is presented in Figure 5.

The *freestanding triboelectric-layer mode* consists of symmetrical electrodes connected by an external load placed under the movable dielectric layer. As the dielectric material changes its position between the two electrodes, the potential distribution changes as electrons pass back and forth between electrodes, resulting in power output. The sliding of an electrostatic charge along the surface of a dielectric material can cause friction in the flow of charge, which eliminates the risk of material surface wear caused by friction and increases its durability [69]. The working mechanism of the freestanding triboelectric-layer mode based on the work of Wang et al. work [78] is presented in Figure 6.

There are many different divisions in terms of the materials used and the complexity of the TENG structure, and we believe that this topic has been comprehensively described in many review papers [79,80,81,82,83]. Therefore, due to the extensiveness of the topic, it will be described very briefly.

Based on the vertical contact–separation TENG, its structure is composed of four different layers: the charge-generating layers (positive and negative triboelectric materials), the charge-trapping layer and the charge-collecting layers (positive and negative electrodes) and charge-storage layers [84]. To achieve the highest possible TENG performance, selecting materials with a high electronegativity difference and appropriate microstructural engineering is essential to maximize efficient surface charge generation [85]. Commonly used substrate materials include natural materials such as silk [86], cotton [87] and paper [88]; polymers, namely polytetrafluoroethylene (PTFE) [89,90,91], poly(vinyl alcohol) (PVA) [92], fluorinated ethylene propylene (FEP) [93], polyethylene terephthalate (PET) [94,95,96] and polydimethylsiloxane (PDMS) [96,97]; metals such as Copper [98] and Aluminum [99]; semi-conductors such as silicon [100,101] or titanium dioxide [102,103] and textiles (cotton [104]). As for conductive material, carbon nanotubes (CNTs), carbon particles, metals and metal oxides, graphene, nanowire-based materials, and conductive polymers have been utilized for their flexible and stretchable characteristics [105].

To design TENG into wearable form with a high softness, flexibility/stretchability, washability and biocompatibility, different approaches have been made in the structural design. Depending on the architecture, TENG developed into configurations such as the following:(a)Kirigami/origami structures—a kirigami structure is created during “folding” and “cutting”, whereas an origami structure is created by “folding” of the material (paper- or plastic-based) [24]. For example, Qi et al. [106] developed a kirigami-inspired TENG where the friction layer was processed into the kirigami configuration with one or two degrees of freedom on a PET sheet by laser cutting technology. The whole structure consisted of PET, Copper, FEP, Acrylic and Sponge materials. This TENG aimed to work as a self-powered acceleration sensor, which can monitor acceleration changes (1–9 m/s^2^) and harvest ultra-wide-band vibration energy (2–49 Hz).(b)Textile structures—fibre form, yarn form (converted from fibres) or fabric form (woven or knitted yarn)—1D, 2D or 3D structures that are easy to fabricate at low cost. They are designed to fit with the user’s body while providing functional properties and comfort [24]. Schematic examples of textile-based generators are shown in Figure 7 [79]. The 1D materials fabricated by electrospinning or surface deposition methods are represented by metallic coated yarns and nanotubes which act as stretchable electrodes. The 2D materials in TENG technology represented by MXenes structures acting as a negative triboelectric layer are a promising approach for boosting the power output of TENG [107]. The 3D materials are represented by fabrics made from fibres converted into yarns, knitted, or woven into fabrics. Fibre-based TENGs are designed as a coaxial structure consisting of synthetic polymer fibres acting as triboelectric material (also as carrier and encapsulation layers) and conductive wires (natural or synthetic fibres) acting as electrodes [108]. They have been discussed in detail by Bulathsinghala et al. [24] and Pen et al. [79].(c)Core-spun/coated textile-based TENGs—among the most complex types of textile-based triboelectric nanogenerators, where multiple layers are integrated into a single material. They typically consist of the conductive layer spun or coated in a thin layer and further spun or coated to create a triboelectric layer in the form of fibre or yarn. Even though compared to other textile-based TENGs their fabrication process is complicated and the energy output is one of the lowest, their advantage is their small size, which helps with integration into clothes [84]. A critical review on the core-spun yarn-based TENGs has been released recently [109].(d)Electronic skin—wearable thin films and thin sheets or nonwoven composites. As for the flexible thin films or sheets, material combinations are used—PDMS/Cu, PDMS/PET, PDVF/PDMS or PDMS/Nylon. They ensure good electrical output, but the comfort of the person who wears them depends on the structure thickness. Thus, close contact with the contours of the body is limited. In the case of nonwoven composites, fibre assemblies are bonded together by mechanical, chemical or thermal treatments. TENG nonwoven E-skins are breathable, self-cleaning, antibacterial, and comfortable materials produced at high speed and low cost [24]. Electronic skin-based nanogenerators will be more broadly discussed in the section related to application possibilities.(e)Nonclothing-based shoes—TENGs can be incorporated into a footwear for energy harvesting or sensing purposes. They can be installed above or under the sole, integrated into it or incorporated within the shoe. All of these have been discussed in detail by Dassanayaka et al. [83].(f)Other configurations—sandwich, honeycomb, ball, nanowires, and others [110].

Novel configurations of TENGs were developed to improve kinetic energy conversion. Yan et al. reported a fish gelatin-based hydrogel as a self-powered pressure- and strain-sensor monitoring human real-time motions. Fish gelatin was integrated into polymer networks with silver nanoparticles forming a structure denoted as FG-Ag hydrogel. Then, PDMS was used as the negative triboelectric layer, and copper foil was used as the positive triboelectric layer, which coated the hydrogel surface—an electrode. The TENG exhibited a high maximum open-circuit voltage of 232 V and a short-circuit current of 1.6 μA [111]. Another example of fish-gelatin-based TENG was inspired by leaf surface-microstructure. Fish gelatin worked as a positive friction layer, whereas PDMS worked as a negative friction layer. The maximum open-circuit voltage of 320V and short-circuit current of 0.8 μA were achieved. This structure was used to power electronic devices and charge capacitors and showed self-powered sensing abilities [112].

TENGs exhibit a wide range of features such as flexibility [101,113], stretchability [114,115], washability [116,117], self-healing ability [118,119], shape adaptability [120,121] and moisture resistance [122,123]. Moreover, triboelectric nanogenerators can accumulate various types of energy and produce electric current. They are lightweight and simple in design, and their conversion efficiency and energy density are high. The materials are inexpensive and readily available, but in many cases, they are also biodegradable and biocompatible [124,125,126]. All these advantages enable a wide range of applications for TENGs, such as energy harvesting devices, human–machine interfaces (HMIs), wearable and implantable electronic devices, self-powered sensors, and others [16]. However, since the energy converted by TENGs is low alternating current (AC), the challenge is to drive electronic devices that require direct current (DC) directly. Therefore, to create a complex energy harvesting system, an energy management module and an energy storage module were introduced and successfully integrated into the triboelectric nanogenerator.

#### 2.2.2. TENG Energy Harvesting Device System

Most TENG-based energy harvesting systems intended to power small electronic wearable/implantable devices consist of three parts—an energy harvesting module (TENG), an energy management module (a rectifier bridge [127]), and an energy storage module. Without a rectifier, there is a significant impedance mismatch between energy harvesting and storage, causing severe power loss and low energy conversion efficiency [72]. When the entire system is subjected to an external load, positive and negative triboelectric charges will appear on the surface of the electrodes due to the triboelectric effect. When the charges are removed, the opposite charges separate and current flows from the top to the bottom electrode due to the potential difference. The energy storage module—a battery or supercapacitor—will start charging until the external force is released, and the triboelectric charges shield the induced potential completely. Applying an external force to the system again leads to a reduction in the existing gap between the triboelectric friction layers. Then, to maintain the balance of the electrode potential difference, the current will flow in the opposite direction. With the help of a rectifier, the energy storage module will be recharged. This entire process transforms the irregular and unstable AC signal into a constant DC output signal, which can then be used to power small electronic devices [128]. The entire TENG-based energy harvesting system is shown in Figure 8 [127].

## 3. Application of Piezoelectric and Triboelectric Nanogenerators

Work on energy harvesting technologies has intensified in recent years due to the increasing number of multifunctional electronic devices appearing in every aspect of our lives. Examples include wearable devices [129], implantable medical devices [130], sensors [131] and other self-powered devices. The common denominator of these devices is their limited lifespan and the fact that they are either worn directly on the user’s body or built into the user’s amenities. As the number of so-called “smart” devices increases daily, providing them with a considerable energy supply becomes a major challenge. Therefore, obtaining energy from human movements and using it as an energy source for these devices is the most feasible and effective technical path [132].

### 3.1. Piezoelectric Nanogenerators

#### 3.1.1. Wearable and Implantable Devices

Piezoelectric energy harvesting devices, like triboelectric energy harvesting devices, can collect a lot of biomechanical energy when attached to various body parts. They use mechanical vibrations caused by body movement to collect this energy and convert it into electrical energy. Both in vivo and in vitro movements can be used to design new self-powered devices. The vast number of piezoelectric biomaterials make them compatible with various environments, especially in contact with the human body, making them excellent candidates for implantable and wearable medical devices, self-powered wireless sensors, and health-monitoring.

#### 3.1.2. Electronic Skin (E-Skin)

Electronic skin, a novel flexible wearable sensor, has been extensively researched, allowing us to receive an ultra-thin, multifunctional device with low energy consumption. Research is currently being conducted to create a new generation of autonomous e-skin for applications in human–machine interaction, virtual reality, and artificial intelligence. The mechanism of electronic skin, based on the piezoelectric effect, is that the pressure applied changes the separation of dipoles in the material, causing electric charges to accumulate on the electrodes. Then, during bending, the output voltage of both sensor layers is synchronized, and the voltage signal is related to the bend radius and angle.

Recently, piezoelectric nanogenerators have been widely used as self-powered sensors. However, their durability constantly faced obstacles related to mechanical damage. Therefore, self-healing properties have been proposed to extend the equipment’s life.

Yang et al. [133] reported a fully self-healing electronic-skin pressure sensor. Piezoelectric lead-zirconate-titanate (PZT) particles and conductive Ag nanowires were dispersed in polydimethylsiloxane (solution), forming both strong dynamic hydrogen bonds that provide high tensile strength and weak hydrogen bonds that contribute to self-healing. A layered structure with a self-healing PZT piezoelectric composite sandwiched between self-healing Ag nanowire electrodes was fabricated to create a fully self-healing piezoelectric nanogenerator (FS-TENG). The working mechanism and output performance for structures with different PZT content (30 wt% to 70 wt%) were investigated. The maximum output voltage and output current at a force of 20 N were observed with a PZT content of 70%—3.2 V and 56.1 nA, respectively. The pressure magnitude and spatial position distribution were correctly identified, depending on the electrical output of the electronic skin. In addition, nine FS-PENG structures (0.5 × 0.5 cm) were fitted together to form arrays of sensors and then secured to the back of the hand to act as an electronic skin. Each system individually generated nine parallel output signals when an external load was applied to the electronic skin. When one of the FS-PENG assemblies was cut in half, there was initially a significant drop in output current. However, the two previously cut-off units reconnected through a self-healing process, and the output current returned to its pre-cutoff value.

In 2023, a self-powered PENG-based design was presented, in which a homogenous layer of PVDF nanofibers prepared by electrospinning as an artificial electronic skin was obtained [134]. The prepared e-skin was mounted on the shoulder, elbow, and knee joints, as well as on the soles of the human and robot feet, which were synchronized (Figure 9). Identification of flexion mode (sensitivity of 0.44 mV/°) and compression (sensitivity of 2.5 mV/N) was possible by detecting various signals from the external pressure. Even after 2500 loading cycles, the electronic skin showed good mechanical flexibility and piezoelectric effect. The accuracy of e-skin tracking was also tested. A grasping activity was evaluated, in which the bending movement of the e-skin installed on a human finger was simulated by a robotic finger with a tracking error of 5%. A slightly higher tracking error of 6% was observed during the gait monitoring process, which correlates with the need for a fall prevention program to ensure the robot maintains an upright posture while walking. These results indicate the prospects for wireless auxiliary sensors.

#### 3.1.3. Textile-Based PENG

Smart textile applications have special requirements, such as stretchability and flexibility, which cannot be achieved using piezoceramics due to their brittleness and high level of hardness; they exhibit excellent piezoelectric and dielectric properties as they are brittle and hard materials. Many attempts, especially between 2017 and 2019, were devoted to creating PENG based on mats, yarns and fabrics that could theoretically be used close to the human body. However, the success rate is relatively low and there are only a few noteworthy examples [135,136,137].

Rafique et al. [138] fabricated a piezoelectric nanogenerator with silver-doped zinc oxide (ZnO) nanorods on cotton fabric as the internal layer and Cu electrodes as the outer layers. The output performance of PENG was investigated for both undoped ZnO nanorods and doped ZnO nanorods and is shown in Figure 10a,b. A mechanical force of 3 kg was periodically applied and the output voltage and output current were obtained. The output voltage and output current values for doped ZnO (6.85 V and 3.42 µA) were almost three times higher than those for undoped ZnO (2.28 V and 1.16 µA). The maximum output power was achieved at 31 MΩ with a value of 1.45 mW/cm^2^, which could be stored for 600 s at an optimal capacitance of 22 nF (1800 cycles).

A year later, a textile-based nanogenerator was produced [139]. The entire structure (PEDOT:PSS/CuSCN/ZnO) consisted of two electrodes—a gold electrode on the top of the PEDOT-PSS and Cu/Ni coated textile as the bottom electrode, and CuSCN/ZnO nanorods in the middle (Figure 11c). Very interesting results were obtained during performance characterization (Figure 12c,d). As the length of the ZnO nanorods increased, an increase in the output voltage and power density was observed. The output voltage values increased from 0.2 V to 1.81 V when the oscillation frequency was increased from 19 Hz to 26 Hz. A maximum power density of 0.38 µW/cm^2^ at 7 MΩ load and an oscillation frequency of 26 Hz was sufficient to power a commercial LCD screen, which offers prospects for portable and wearable self-powered textile-based electronic devices.

#### 3.1.4. Self-Powered Sensors

The piezoelectric effect is a popular method for developing sensors to measure acceleration, acoustic force, vibration, force and load, pressure and strain, and more. As a result, special attention has been paid to physical sensors such as pressure, force, voltage, and force sensors [140,141,142]. However, attention has now turned to new areas, such as self-powered wireless and signal-transmission devices.

In 2022, a metal-free N-methyl-N′-diazabicyclo[2.2.2]octonium-ammonium triidodide perovskite PENG (MDABCO-NH_4_I_3_; MN-PENG) was produced for the first time [143]. The output performance of the perovskite PENG was tested, and under a 0.55% strain and a frequency of 3 Hz, the output voltage and output current reached 15.9 V and 54.5 nA, respectively, suggesting potential application as a self-powered strain sensor with excellent stability and no significant degradation, even after 5000 bending cycles. MN-PENG was applied to light-green LEDs and to charge a 4.7 µF capacitor (in 7 min—4.8 V). Various body movements—wrist, elbow, biceps, and neck—were detected, and a signal was provided for real-time interactions between human and machine. Moreover, five MN-PENGs were installed on each finger of the hand, and the gestures assigned to the corresponding fingers were immediately displayed by obtaining output voltage (visualized as symbols on the computer) and output current signals (Figure 13b,c). These findings shed new light on using perovskite-based PENG for biomedical and self-powered sensor applications.

One late development was a self-powered PENG sensor composed of poly(vinylidene fluoride-co-trifluotoethylene) piezoelectric nanofibers and carbon nanotube electrodes, which were then encapsulated using initiated chemical vapour deposition to form nano-coating. This method gave the sensor superhydrophobicity, self-cleaning ability, and an antifouling effect against Gram-negative *E. coli*. Gram-positive *S. aureus* bacteria was also confirmed at the level of 90%. The results showed that the output voltage remained close to 60 V at 90% relative humidity. High sensitivity was also confirmed during gait analysis (Figure 14), where sensors were integrated into various parts of the shoe inserts. All detected output signals can be successfully assigned to one of eight different sensor positions, with negative and positive peaks corresponding to foot sensor contact–separation cycles (Figure 15b,e) [144].

Another invention based on a PVDF-derivative, namely poly(vinylidene fluoride-co-tetrafluoroethylene)—P(VDF-TrFE)—was introduced as a transparent self-powered PENG force sensor [145]. P(VDF-TrFE) film was placed between indium tin oxide electrodes to create a 2 × 2 PENG array. The output voltage reached a maximum at 10 MΩ load. When a 4 N force was applied to the PENG, an increase in the output voltage was observed, corresponding to the rise in the frequency. The voltage increased from 0.6 V at 2 Hz to 2.7 V at 10 Hz. This frequency–piezoelectric output increase dependency could be used to measure mechanical vibrations, orientations, and accelerations.

#### 3.1.5. Self-Powered Implantable Electronics

The search for new biocompatible and biodegradable materials has brought new potential for in vivo applications in implantable piezoelectric nanogenerators. They have proven to be particularly useful for extracting energy from the movements of body organs. In terms of advantages, implantable PENGs have high overall accuracy, generate little or no heat during operation, and have a vast selection of ceramics and polymers with high power output, which is an excellent start to developing self-powered devices [146]. Disadvantages include the toxicity of lead-containing piezoelectric materials or the sensitivity of some materials to body fluids; they are unable to generate energy in a static state, and they are unable to function if there is no charge polarity in the piezoelectric materials [147].

A very interesting example was presented by Park et al.—a “seamless human oral motion-powered dental implant system called Smart Dental Implant or SDI” (Figure 16) [148], in which a piezoelectric dental crown was able to collect energy through chewing and teeth-brushing movements, and then used for photo-biomodulation therapy (PBM). Oral mechanical motions generated energy temporarily stored in a 47 µF capacitor with embedded illumination LEDs. Depending on chewing force, the average voltage outputs ranged from 0.4 to 1.3 V; in brushing motions, the voltage outputs ranged from 0.7 to 1.0 V. The mechanical strength of dental implants was comparable to that reported in other works—in the mentioned paper, the flexural strength was 50 MPa. The flexural modulus was 6630 MPa, while in [149], the flexural strength was 105 MPa, and the flexural modulus was 2840 MPa, and in [150] it was 130 MPa and 1200 MPa, respectively. The data confirm sufficient mechanical strength suitable for dental implant materials.

In 2019, researchers from China [151] demonstrated a high-performance piezoelectric energy generator iPENG implanted in an adult porcine pericardium to extract energy from the heartbeat and directly power a commercial cardiac pacemaker. The entire structure consists of three main parts—an encapsulation layer, a piezoelectric composite layer, and a flexible skeleton layer. In vivo studies were performed by implanting four different pericardial sites: the apex and three walls—anterior, posterior, and lateral (Figure 17a). The maximum output voltage and short-circuit current were recorded for apex site implantation and were 20 V and 8 µA in series mode and 12 V and 15 µA in parallel mode, respectively. The developed iPEG pacemaker produced a pacing pulse of 2.5 V amplitude at 80 beats per minute, comparable to a battery-powered pacemaker. This implantable energy generator strategy is an excellent example of future development, especially in myocardial stimulation and cardiac therapy.

A similar invention was introduced three years later by Dong et al. [152], who developed a self-powered universal pacemaker. The structure consists of a PDMS layer, Silver nanowires and potassium-sodium niobate piezoelectric particles (KNN; (Na_0.52_K_0.44_Li_0.04_)(Nb_0.88_Ta_0.12_)O_3_)). To evaluate the structural design, 1 × 2 cm^2^ size nanogenerators were prepared. A force of 2 kPa (frequency of 1 Hz) was applied from a linear motor to measure the output characteristic of the PENG device. An output voltage of 98 V and a short-circuit current of 3.2 µA were measured. The maximum power density reached 22.5 µW/cm^2^ (with an external loading resistance of 10 MΩ). Moreover, the 2 × 6 cm^2^ structures were tuned to collect biomechanical energy during hand bending and head nodding, generating 13 V and 15 V, respectively. The energy generated during hand slapping was enough to be stored in two commercial capacitors (2.2 and 4.4 µF) to power 4 LEDs. The PENG device was also implanted in a canine model to evaluate the feasibility of extracting heartbeat energy for cardiac pacing. The structure consisted of a PENG device, a rectifier bridge, a capacitor, a reed switch, and a wireless trigger. Three different pacing sites—right atrium, left ventricle, and bundle of His—were experimentally tested (Figure 18). Multiple pacing sites (in both single- and dual-chamber pacing modes) were verified during epicardium ECG recording. After a minute of accumulating the energy of the heart’s pulsations, the reed switch responsible for releasing electrical stimuli was triggered to pace the heart. This proves the validity of the concept and the possibility of its implementation in the human body in the future.

Another very interesting example was proposed by Fan et al. [153], where researchers investigated the possibility of harvesting energy during mastication. An unimorphic piezoelectric nanogenerator composed of a macro fibre composite and a titanium substrate was created and installed on a synthetic mandible based on finite element analysis to ensure the best biocompatibility and structural flexibility and the closest possible reflection of a real human mandible. A loading device that mimics chewing forces has been developed. Tests have shown a peak-to-peak voltage of 1 V and an average power of up to 1.27 µW, which can be amplified using a multi-layer microfiber composite structure This is a promising way to produce energy that could power deep-brain stimulation devices in the future.

#### 3.1.6. Triboelectric Nanogenerators

##### Wearable and Implantable Devices

A wearable device is a small, portable electronic and software-controlled product that can be worn directly on the user’s body or carried with them. Depending on the specific function, it may function as an independent work unit or work indirectly through integration into clothing, footwear, and other wearable accessories. Wearable TENG devices are highly customizable, stretchable, and flexible, which has led to a wide range of applications, from electronics such as smartwatches and smart glasses, smart shoes and smart clothing, to electronic skin, biomedical monitors and human–machine interface [154].

##### Electronic Skin (E-Skin)

One of the main categories of wearable TENGs are textile structures (mainly fibres, fabrics, and yarns) and electronic skin (e-skin).

An electronic skin is an electronic device capable of transmitting mechanical signal energy into electrical energy. The device typically consists of a power supply, sensors and actuators, and signal acquisition and processing paths. E-skin has been used predominantly as sensors in wearable, implantable interventional medicine. Electronic-skin technologies enable continuous tracking of human body parameters such as oxygen and blood flow, blood pressure and heart rate, body temperature and other physiological signals. In addition, indicators showing intraocular pressure and blood glucose levels can be detected, which provide real-time information on glaucoma or diabetic diseases. Since the use of e-skin is closely related to the medical industry, some limitations and requirements must be met, such as lightness, flexibility and stretchability, biodegradability, and biocompatibility with the human body [155]. The triboelectric nanogenerator-based electronic skin uses passive–active sensing (contact electrification and electrostatic induction) to transform mechanical signals and generate electrical signals. However, continuous energy harvesting is difficult, and the overall structure is complex. Despite these problems, recent research shows significant progress in the field.

Cai et al. [156] designed a triboelectric nanogenerator based on double-crosslinked PDMS (DCS-TENG) as an electronic skin that exhibits self-healing, shape-adaptive and super-stretchable properties. The triboelectric layer was prepared using polydimethylsiloxane (PDMS), and the ratio of imine bonds to hydrogen bonds was adjusted. In contrast, the hydrogen bonds formed between MXene and PDMS prepared the electrode layer. The power density, output voltage and current were measured while loaded with external resistors, as shown in Figure 19. A maximum power density of 0.98 W/m^2^ was achieved with an external load resistance of 100 MΩ, which allowed more than 80 LEDs to be lit (Figure b in Figure 19 Right). The self-healing function enabled the conversion of individual sensors into a sensor consisting of four identical and individually operating sensors.

Liu et al. [157] reported a highly transparent (over 92%) and healing triboelectric e-skin with a thickness of 3 μm, based on a uniformly separated aliphatic disulfide-reinforced microphase elastomer that activates the self-healing function (Figure 20). The device comprised a triboelectric layer (elastomer) and an electrode (PEDOT:PSS). The device exhibited a maximum output power of 965 nW/cm^2^ at an external load of 20 MΩ. The open-circuit voltage, current density and charge density were 26 V, 200 nA/cm^2^ and 12 nC/cm^2^, respectively, indicating energy harvesting capability. Furthermore, a plastic screen protector was laminated onto the skin and then attached to the mobile phone screen (Figure 21). Waving the hand over the screen at a distance of 1 cm to 10 cm generated a voltage of 0.11 to 0.08 V, respectively. During these movements over specific distances, the external circuit will induce and capture voltage. The triggered signal will be activated and sent to the corresponding application installed on the smartphone, and, after processing the signal, it is possible to control the smartphone, i.e., answer or end a call using touchless gestures, which is new to this topic.

Another interesting work was presented by Lu et al. [158], who proposed a prototype of a lip-language decoding system equipped with self-powered flexible triboelectric sensors operating in contact–separation mode. In addition to the sensors made of copper-plated polyvinyl chloride and polyamide films (Figure 22a), the system consists of fixing masks that help locate TENG at specific positions of the lip muscles, an electronic reading system, and neural-network classifiers (Figure 22b). Electrical signals are generated when you speak, and the TENG sensor detects lip movement (Figure 23). Electrical signals are read, processed, and sent to a neural network, where the information is reproduced as sound or written as text on the screen.

#### 3.1.7. Textile-Based TENG

Textile-based TENG devices are an evolution of primary TENG devices with an optimized structure suitable for many practical applications requiring durability. Textile-based TENG structures are popular because they can be integrated or sewn into clothing. Textile properties such as ease of wear, breathability, elasticity, and stretchability allow the measurement of a wide range of body movements. Their development introduced properties such as washability and better moisture removal. What is more, when designing the device, visual features and aesthetics were also taken into account, as the appearance of clothing significantly impacts consumption. Since textile-based TENGs come into contact with human skin, various tactile-comfort properties are of great importance, namely the absence of skin irritation or discomfort upon contact, thermal comfort and static electricity. Materials must be non-toxic, biocompatible, and safe in the long term. Textile TENG materials are typically used in three forms—fibre, yarn, or fabric. Textile fibres can be natural or artificial. By spinning or extruding the fibres, a yarn can be obtained. Various types of fabrics can then be produced using knitting or weaving processes. In addition, new techniques such as electrospinning, electrospraying, dip-coating or 3D printing have been used to enrich the manufacturing process [84].

Paosangthong et al. [159] designed a novel textile-based TENG with positively and negatively woven polytetrafluoroethylene vinyl fabric and nylon fabric strips, operating in a freestanding-layer mode (Figure 24a). The woven TENG was attached to the lower part of the sleeve and the bottom electrodes (polyester fabric coated with Ag) were placed in the lab coat at two different parts of the body—the hip and the waist—to collect the energy of human movements (Figure 25a). An open-circuit voltage of approximately 0.75 Hz and 2.30 Hz was generated by arm swing during walking and running, respectively. Short-circuit currents of nearly 120 V were detected for both activities. The output energy generated from walking and running was then used to charge a 4.7 µF capacitor, which was charged to 2.2 V and 5.2 V after 30 s. This energy level was sufficient to power three different electronics—a “STOP” sign, a digital watch and a Bluetooth transceiver. The novel use of positive and negative triboelectric material in a single structure increased the generated power by 2.2 times compared to other TENG devices based on a single triboelectric material structure.

Gao et al. [160] designed a TENG based on core-spun coating yarn, in which silver-plated nylon yarn acted as the core and electrode, and cotton fibres served as the surface and base material for coating with triboelectric materials. Braiding technology was used to realize a hierarchical structure of multiple core-spun yarns covered with nylon and the addition of polydimethylsiloxane acting as positive and negative triboelectric materials. After five hand-washing cycles, the TENG demonstrated washability and 174 V output voltage stability. The peak power density reached a maximum of 275 mW/m^2^ with a load resistance of 50 MΩ. The output power generated by the device was enough to light 200 LEDs (Figure 26b). By designing energy-harvesting shoe inserts, the device harvested and converted energy that powered more than 78 LEDs while walking. Moreover, the charging capacity was investigated and it was shown that the energy generated by the TENG could be directly used to drive small electronic devices such as an electronic watch or an electronic calculator (Figure 26d).

Another interesting example was developed by Wen et al. [161], where a polyester fabric was transformed into an energy-harvesting and self-powered human–machine interface (HMI) glove with human-motion-detection properties (Figure 27). The superhydrophobic material absorbs the energy when you bend your elbows, tap your hand, and walk or run at a high relative humidity of 76%. The power density of biomechanical energy collected during human activities was almost four times higher than that of unmodified material (0.18 W/m^2^ vs. 0.05 W/m^2^). Although each finger had only one triboelectric sensor, the glove-based HMI could recognize and distinguish between complex and similar gestures. In addition, using carbon nanotubes and a thermoplastic elastomer as a coating resulted in a superhydrophobic textile that minimized the negative effect of human sweat, which also improved recognition accuracy.

#### 3.1.8. Self-Powered Sensors

Sensors are devices designed to detect signals emitted in the surrounding environment and transmit this information to other devices, most often a computer processor. There are two categories of sensors—active and passive. Active sensors, such as temperature sensors, accelerometers or gyroscopes require power sources such as batteries or other energy storage devices, while passive sensors do not require any additional power supply [162].

Nowadays, smart sensors have been successfully incorporated into our daily lives as a health monitoring system—for example, blood pressure or blood sugar level monitoring, environmental protection, infrastructure detection and protection, etc. With the rapid development of the Internet of Things, sensor networks and microelectromechanical system technologies with a considerable number of miniature sensors connected to a larger structure that acts as a data collection layer, the power demand for these sensors is also increasing [125]. In this case, self-powered sensors based on triboelectric nanogenerators were investigated and developed as touch and pressure sensors, motion sensors, chemical sensors, and acoustic sensors. TENG-based sensor units do not require an external power source to generate the electrical signal because the TENG itself produces the output signal. Regarding the energy produced by human movements, tactile and pressure sensors have been studied extensively. The output signal and the touch applied to the TENG structure can be measured, as can information such as pulse amplitude, polarity, periodicity, and waveform. Contact pressure can also be detected and measured from the output signal. What is more, by increasing the contact area between the surfaces, the output amplitude of the TENG can be increased. By increasing the contact pressure, the output level is also increased [127].

One of the primary sources of energy needed for human movement is the activity of the feet. Zhang et al. [163] developed a textile-based TENG in smart socks that transmitted wireless sensory information while collecting energy from body movements (Figure 28a). A mm-scale frustum structure was patterned on the silicone rubber surface, which enabled better gait detection. A power of 3.18 mW was measured when operating at a frequency of 2 Hz. This sock charged a 27 µF capacitor in 4 min, which, theoretically, could support a Bluetooth sensor to transmit data such as temperature and humidity. The accuracy of detecting predefined motions for a specified user reached 96.67%. Additionally, it can be used in real-time VR projection, where the user’s movements and position can be tracked and projected into the virtual space (Figure 29 and Figure 30b). This idea can be utilized with various functionalities in homes, classrooms, work-related spaces, healthcare facilities, etc.

In 2022, Feng et al. [118] reported a self-healing, self-powered sensor based on eutectoid TENG. The electrode consisted of eutectoid-based sulfonated lignin and Fe^3+^. After immersion in a deep eutectic solvent, an eutectogel with properties such as high stretchability of up to 450%, transparency of 93.5% and ionic conductivity of 8.70 mS/cm was obtained, which was stable at temperatures from −80 °C to 25 °C, which indicates antifreeze properties. The effectiveness of mechanical self-healing was up to 96% after 6 h of healing, and the healed materials withstood stretching up to 700% without damage. To verify the self-healing properties, a system consisting of a capacitor, a rectifying bridge and an electronic watch as an external load was constructed. Tapping an eutectogel TENG for 54 s at a frequency of 3 Hz produced 3.1 V energy for a 22 µF capacitor, which powered the electronic watch for 20 s (Figure 31c). An eutectogel TENG sensor operating in single-electrode mode was also built to observe output voltage signals corresponding to different motion frequencies (finger, wrist, arm, and knee-flexion movements) and to prove its suitability as a human-motion monitoring system (Figure 32d,e).

In the same year, Dong et al. [119] fabricated a stretchable and conductive hydrogel-based TENG for energy harvesting and motion detection. A sensing material consisting of polyacrylamide/poly(acrylic acid)/graphene/PEDOT:PSS hydrogel was sandwiched between carbon nanotubes/poly(dimethylsiloxane) films. The mechanical properties showed excellent elasticity (almost 500% stretchability) and a high recovery rate of up to 66%. The hydrogel also exhibited the ability to self-regenerate. The effectiveness of self-healing after stress and strain reached 76% and 85%. These parameters enabled the hydrogel encapsulation in carbon nanotubes/poly(dimethylsiloxane) films to create a system capable of detecting compressive movements and monitoring knuckle, wrist, elbow, and knee-bending movements. Then, the charging capability was evaluated by combining the deformable TENG with a capacitor, LEDs, and a hygrometer thermometer (Figure 33a). The capacitor was charged to 6 V within 4.5 min, which allowed the hydrometer thermometer to be powered. Moreover, the device lit 52 yellow LEDs simultaneously.

#### 3.1.9. Self-Powered Implantable Electronics

An ageing society and a constantly growing number of sick and disabled people mean that the demand for wearable and implantable medical devices is related to diagnostics and monitoring of human activity, treatment and drug delivery, and rehabilitation and surgery. Healthcare technology, including medical devices, urgently needs solutions that enable implantable micro- and nano-systems to function in the body for as long as possible, without endangering human health and life. This is possible by designing devices that harvest the body’s biomechanical energy. Triboelectric nanogenerators have sensing and power functions, making them suitable for in vivo biomedical applications. This allows us to obtain physiological information such as heart rate, pulse, respiratory rate, etc. Depending on their operating time, they can be divided into durable TENG for long-term processing with a stable structure and degradable TENG for short-term treatment with a transient structure [154].

An example of a robust TENG-based device is the pacemaker, which has become a fashionable invention, especially after the COVID-19 pandemic, which caused an increase in the number of cases of sinusitis, atrioventricular block and atrial fibrillation [164]. Ryu et al. [165] reported an inertia-driven TENG device with a coin-sized battery. A power-management integrated circuit and commercially available (Li)-ion battery was integrated into the pacemaker to create a self-powered system (Figure 34c). In vivo evaluation of the TENG device was performed by implanting it on the back of an adult mutt. The device drew more than 140 mW and converted biomechanical and inertial energy into electrical energy at a 4.9 μW_RMS_/cm^3^ rate. This work, in which, compared to previous publications, an indirect supply of energy to the inertia-based TENG was achieved, made it possible to enclose the entire system within the body, for it to function as a medical implantable device. Another robust TENG-based application has been developed as a cardiovascular system-monitoring–endocardial-pressure sensor.

Liu et al. [166] introduced a miniaturized, self-powered endocardial pressure sensor implanted into a porcine model heart (Figure 35a). This ultrasensitive and flexible TENG-based device is designed to convert the energy generated by blood flow inside the heart chambers into electrical energy. The device consisted of two triboelectric layers (nano-PTFE film and Al foil), two electrode layers (50 nm gold layer and Al foil), a spacer layer inserted between the triboelectric layers (made of ethylene-vinyl acetate co-polymer) and an encapsulation layer (Kapton/nano-PTFE layer). The device demonstrated remarkable stability in a humid environment after 100 million swinging cycles. Electrical outputs and changes in endocardial pressure allowed the detection of ventricular fibrillation and ventricular premature contraction (Figure 35d). This work remains one of the best examples of a real-time analysis device for diagnosing and monitoring cardiovascular diseases.

Transient electronics, which consist of degradable materials, are used for in vivo applications such as intelligent monitoring of physiological signals and intelligent therapeutic and disease treatment devices. Special attention is paid to biocompatible nanogenerators with biodegradable and bioresorbable properties that can be used as temporary bio-implants, self-powered electronics, and sensors [167]. Biodegradability ensures that a device that has completed its working cycles can be disposed of without surgery [155].

In 2021, Ouyang et al. [168] introduced a bioresorbable sensor (BTS) based on TENG, consisting of two triboelectric layers—a nanostructured PLA/C film and a Mg-coated nano-structured PLA/C film; and two electrodes—a nanostructured PLA/C film coated with magnesium, and poly(1,8-octane diol-co-citric acid) as an adhesive layer. The BTS sensor was tested in a dog by installing it on a vascular wall to obtain an ambulatory blood pressure signal comparable to commercially available blood pressure sensors. The BTS sensor achieved a 5-day service life and an absorption time of about 84 days, a high service efficiency of almost 6%, a sensitivity of 11 mV mmHg^−1^, good linearity of over 99%, and a stable output performance (2 V) that remained constant under the influence of 450,000 mechanical stimuli. Thanks to a structure composed of bioresorbable materials such as poly(lactic acid) and chitosan, the BTS sensor achieved 99% sterilization. It is envisaged that such an invention will be used as a bioresorbable electronic device in the postoperative period.

Li et al. [169] presented a biodegradable sodium alginate-based TENG operating in a single-electrode mode with antibacterial properties for energy harvesting and self-powered sensing. Composite layers of sodium alginate and glycerol served as the triboelectric layer, and patterned conductive sodium alginate/Ag nanowires acted as electrodes. High-pressure sensitivity (0.237 V/kPa under 3 kPa pressure) influenced the production of a multiple-point tactile sensor that detected real-time wrist, finger, forehead, and throat muscle movements. The transparency and flexibility of the sodium alginate film enabled the development of a self-powered human–machine interface system consisting of a 3 × 3 TENGs matrix, a signal processing circuit, and a computer, in which the generated voltage signals were used to control the movements of characters in a game played on the computer (Figure 36h–j). During testing, an output voltage of 53 V, a transferred charge of 18 nC, and a peak power of 4 µW were achieved.

## 4. Hybrid Systems

Both triboelectric and piezoelectric nanogenerators exhibit a wide range of properties and applications. Triboelectric nanogenerators are characterized by high efficiency, easy construction, lightness, and high flexibility. However, in many cases, using a single triboelectric nanogenerator will not be enough to drive small- or medium-sized electronic systems. Therefore, a conjunction strategy was proposed—the creation of hybrids (Figure 37). There are two ways to achieve hybridization of energy harvesting devices—either by integrating different energy harvesters or by combining available energy conversion mechanisms from different sources [170]. The integration of various energy harvesters involves harvesting energy from the environment: solar, thermal, or mechanical energy. Triboelectric and piezoelectric energy harvesters are often integrated with other energy harvesters due to their mechanical flexibility and design diversity. However, the combination of the triboelectric and piezoelectric effects increases the charge density and the magnitude of the output current. There are many examples of the combination of triboelectric nanogenerator (TENG) and piezoelectric nanogenerator (PENG) [171].

Liu et al. [172] developed a piezoelectric-triboelectric hybrid nanogenerator based on an interconnected lead-free and flexible composite film made by encapsulating a composite of bismuth ferrite (BFO) and glass fibre fabric (GFF) with polydimethylsiloxane (PDMS), where the BFO-GFF/PDMS structure acted as both a piezoelectric layer and a triboelectric layer. The results showed that the highest possible short-circuit current density of 3.67 µA/cm^2^ and open-circuit voltage of 115.22 V was reached at 1 Hz during contact–separation movements. In contrast, the power density reached the maximum at a load resistance of 250 MΩ, reaching 151.42 μW/cm. Converting human movement into electrical energy was possible by moving a device mounted on the arm while changing posture (Figure 38b,c). When bending and extending the arm, an output current of up to 1.5 μA was generated. Moreover, the high sensitivity of the BFO-GFF/PDMS film enabled the device to capture infinitesimally small movements such as clenching and opening a fist. Furthermore, a hybrid with a contact area as small as 2 × 3 cm^2^ was successfully utilized as a power source to illuminate three red LEDs without any external or additional storage units.

Du et al. [173] reported a hybrid shoe insole in which a sandwiched triboelectric nanogenerator structure forms the front part, and an arched piezoelectric nanogenerator forms the back part. The TENG part consisted of three layers of PTFE-Al-PTFE and the PENG part consisted of a thick PVDF film covered with a PE film and bonded on the bottom with Kapton film. Moreover, two sensors (a hybrid insole nanogenerator and dorsalis pedis artery) were integrated into a self-powered dorsalis pedis artery system intended for medical monitoring and a health warning system. The device achieved maximum open-circuit voltage, short-circuit current and amount of charge during jumping, of 150 V, 4.5 µA and 240 nC, respectively. The maximum output power was achieved during stepping under a 40 MΩ workload, reaching 77 µW. Also, the 100 µF capacitor was charged in 8 min to 2.5 V and a 47 µF capacitor was charged to 1.7 V in 150 s, effectively powering the calculator for 25 s. In addition, the hybrid nanogenerator directly powered 60 LEDs.

Chung et al. [174] designed an origami-based piezoelectric–triboelectric hybrid generator. The structure consisted of two outer layers—TENG working in the vertical contact–separation mode, a rotational TENG, and an inner PENG layer. The origami shape consisted of a triangular cylinder that formed a 3D shape. Under vertical pressure loading, all nanogenerators were able to generate electricity separately. Through continuously applying and removing external pressure, the rotary TENG produced an open-circuit voltage of 15 V. For the combined structure, the peak for open-circuit voltage was 120 V and for the closed-circuit current it was 90 µA. The hybrid design successfully charged a commercially available 22 µF capacitor, which, under compression, lit 60 LEDs.

## 5. Comparison between Piezoelectric Nanogenerators and Triboelectric Nanogenerators

Both the piezoelectric nanogenerator and the triboelectric nanogenerator have individual, unique properties that distinguish them from other technologies. There are some similarities, such as generating alternative-current-based electrical outputs or similar impedances. However, there are many differences in materials, manufacturing process and costs, power output levels, mechanical performance, and more. In the case of PENG, materials from the ceramic group are brittle and require complex manufacturing steps, and the cost of an individual piece is also higher than in the case of TENG. Some require an additional encapsulation step to minimize the possible toxic effect of lead-based materials. The mechanical performance is also poor, which causes the problem of the limited deformation that can be applied. Piezoelectric energy harvesting works best when harvesting muscle movement—arm, leg, feet, finger, and palm grasping. They are installed in a backpack, on the wrist, or in footwear, to collect mechanical energy. However, an upgrade is needed to capture very low-frequency vibrations and low-frequency human movements for better performance. Thus, techniques such as the double pendulum system, frequency conversion and circuit management have been implemented for effective energy harvesting [53]. Concerning nanogenerators as a power source, the main direction of development for PENG structures is high output power, flexibility, and lack of irritation during contact with the skin, as it is used for both wearable and implantable devices [16].

Conversely, TENG has the advantage of many materials due to the widespread occurrence of the triboelectric effect. The unit cost is lower and the compatibility is better than PENG. The functional area of TENG can also be increased, resulting in a higher output density. Designing miniaturized devices and power circuits is simple. The mechanical stability is very high, allowing the TENG device to withstand up to millions of cyclic impacts with very little damage, which can be reversed if necessary thanks to its self-healing properties. TENGs provide a stable power source that can be applied to human skin, textile-based clothing, for energy harvesting during hand tapping or biomechanical energy harvesting during contraction and relaxation of the heart and lungs [53]. In order to increase the efficiency of energy harvesting methods, such as the in-plane charging–separation mode, the core-shell-structure method and the ultrathin single or liquid-metal electrode have been implemented. However, moisture and dust can negatively affect TENGs [16,19].

For these two effects, it can be concluded that these techniques complement each other, and their combination can increase each of their functionality and overall performance individually. Therefore, the PENG-TENG hybrid system has been successfully designed to enhance the output performance.

A summary of the commonly used materials, structure, output performance, and the advantages and disadvantages of PENGs and TENGs is shown in Table 3. Table 4 summarizes the piezoelectric, triboelectric and hybrid kinetic-energy devices that have deserved attention in recent years (2019–2023).

## 6. Challenges and Future Outlooks

This review presents the latest developments in triboelectric nanogenerators, piezoelectric nanogenerators and their hybrids, using human-derived kinetic and biomechanical energy for applications such as implantable or wearable devices and self-powered sensors. Although nanogenerators are currently a trending technology, based on the research conducted, we conclude that no significant breakthrough has been achieved that could revolutionize the market and industry for self-powered electronic devices, which continue to gain popularity. Existing challenges that still need to be urgently addressed are the following:Power management—although high output voltages have been achieved for many nanogenerators, power density needs to be improved to meet real-time power demands in practical applications. In the case of TENG technology, more effort must be put into improving the charge generation, transfer, and collection, to increase conversion efficiency. This is also closely related to the energy-storage system, which is currently primarily based on batteries (both rechargeable and non-rechargeable) or capacitors [202].Material selection and environmental factors—materials selected for TENG devices must exhibit mechanical flexibility, elasticity, and durability during mechanical deformation. They must adapt to mechanical damage and maintain performance under various motions such as bending, twisting, and stretching. It is crucial to identify suitable pairs of materials that would exhibit high triboelectric properties. For both TENG and PENG, materials should be stable under varying temperature and humidity conditions and exposure to chemicals, moisture (including human sweat) and contaminants, without losing performance in the long-term operation. Moreover, biocompatibility and biodegradability should be considered when manufacturing implantable and wearable devices in direct contact with human organs and skin that can cause inflammation or infection. More natural materials should be tested and chemically or physically modified to promote their properties [202]. The possibility of recycling, and thus creating green and environmentally friendly PENG and TENG devices, is one way to reduce the amount of waste produced during traditional implantable- and wearable-device manufacturing.Design and integration—the device should be lightweight and small, adapting to the shape and movements of the body without hindering it or causing uncomfortable sensations. Integration with textiles and electronic devices requires a well-thought-out view of the entire design, which should combine functionality, comfortability, and aesthetics on one hand, and coexist with necessary and critical parts such as energy storage units, power management and conditioning circuits and other components without interfering with their operation. The cost aspect, however, must be strictly controlled and planned for large-scale production, while maintaining high quality, energy harvesting efficiency and integrity of the overall structure, and at the same time should be available to a broader range of the population [146,202].

Future outlooks:More emphasis should be placed on material selection and structure design to create a system that will operate for a long time without significant performance loss. One idea is to use materials that exhibit self-healing properties as friction layers or fully healable nanogenerators and extend them to large-sized devices [203,204,205,206]. Another idea is to implement 3D-printing method that expands possible application tunability and uses a unique, tailor-made device architecture [207,208,209,210].More designs should be created based on a hybrid system that combines the best features of the nanogenerator, which can be a rigid structure with higher power generation on the one hand and a flexible structure for better wearability on the other. The combined nanogenerators produce more electricity per unit volume or unit area than separately. The problems associated with the significant difference in frequency, amplitude and waveform during energy conversion using different transducing methods should be solved to achieve high energy-conversion efficiency when collecting multiple forms of energy from the environment. One promising development is the combination of robotics and energy harvesting based on hybrid systems [211].Industrialization—currently, most nanogenerators are hand-made prototypes made in the laboratory that only demonstrate one possible application. Therefore, a standardized manufacturing process for different types of nanogenerators on an industrial scale for commercial application in the future needs to be thought of [84,212]. Moreover, most energy harvesters are intended for special applications such as military or medical. More effort must be made to create devices that the average user will wear—for example, a smartwatch integrated with an energy harvester [213].AI, HMI, IoT—with the development of the virtual world, new concepts such as artificial intelligence, the human–machine interface and the Internet of Things have been developed and woven into various aspects of our lives and technology. This is also a very promising way to enrich energy harvesting technology with new application scenarios—wireless signal transmission and multi-dimensional sensing, and visual, auditory, and tactile modes. The devices should be supported by Bluetooth or Wi-Fi communication to monitor and send a real-time signal to the user, for example, regarding their vital signals or position [202].

## 7. Conclusions

In conclusion, significant progress has been made in TENG and PENG technology over the past years. However, as discussed in the previous paragraph, much work still needs to be done. The development of future trends must be adapted to industrial capabilities and to meet many requirements set by future users, which will undoubtedly simplify and improve their lifestyles. In our opinion, this technology can revolutionize various fields—biomedicine, military, wearable and implantable electronic devices, sensors and actuators, material science, and engineering technology—and create a self-sufficient, intelligent system in various environments. Therefore, further research and development of more efficient energy harvesting devices that meet the high energy demand are necessary.

## Figures and Tables

**Figure 1 sensors-24-01069-f001:**
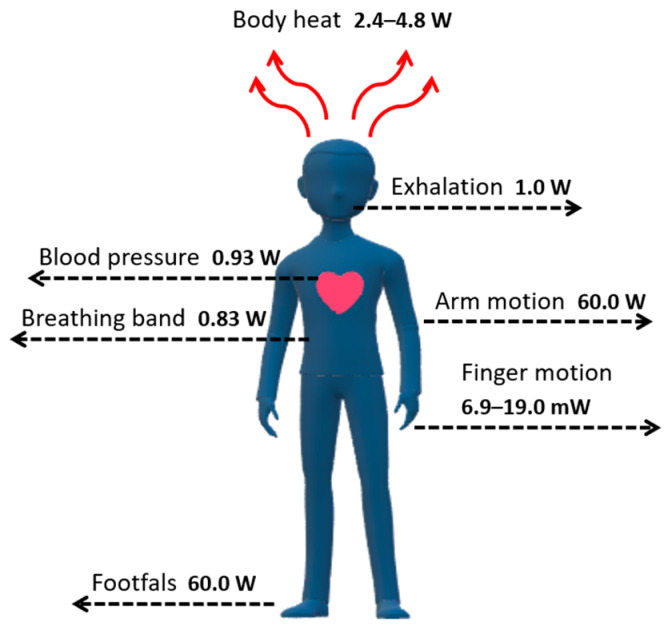
Electrical energy generated by the human body; maximum power for each action.

**Figure 2 sensors-24-01069-f002:**
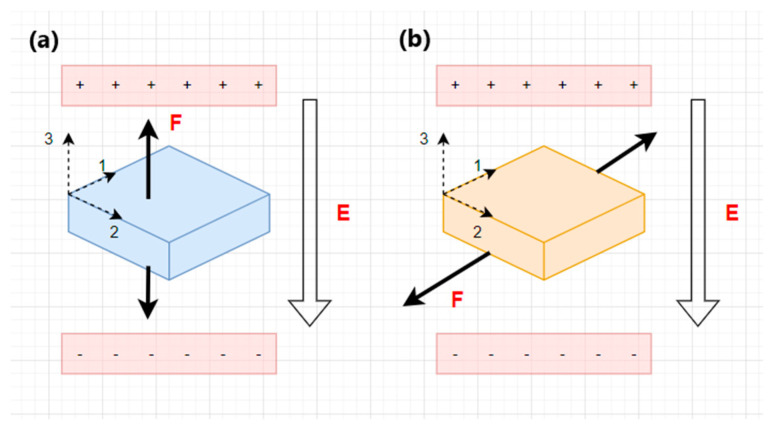
Piezoelectric modes. (**a**) Transverse mode. (**b**) Longitudinal mode. F—the applied force; E—an electric field.

**Figure 3 sensors-24-01069-f003:**
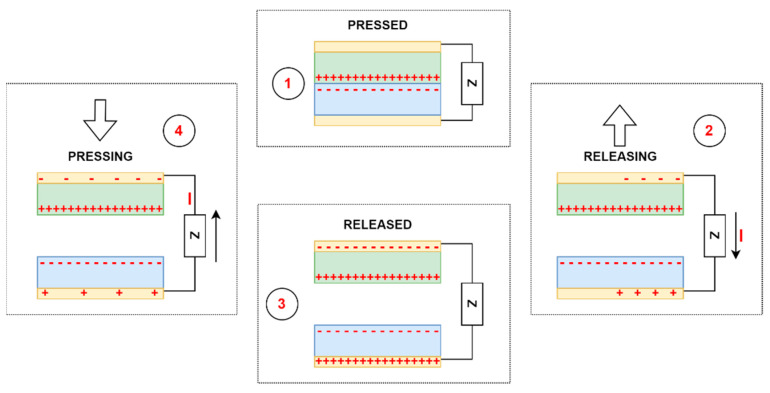
The working mechanism of the vertical contact–separation mode. Running order from 1 to 4. Z—resistor; I—current.

**Figure 4 sensors-24-01069-f004:**
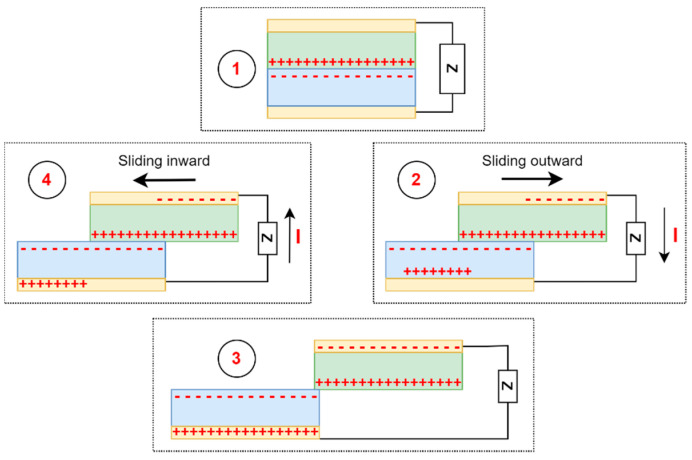
Working mechanism of lateral-sliding mode. Running order from 1 to 4. Z—resistor; I—current.

**Figure 5 sensors-24-01069-f005:**
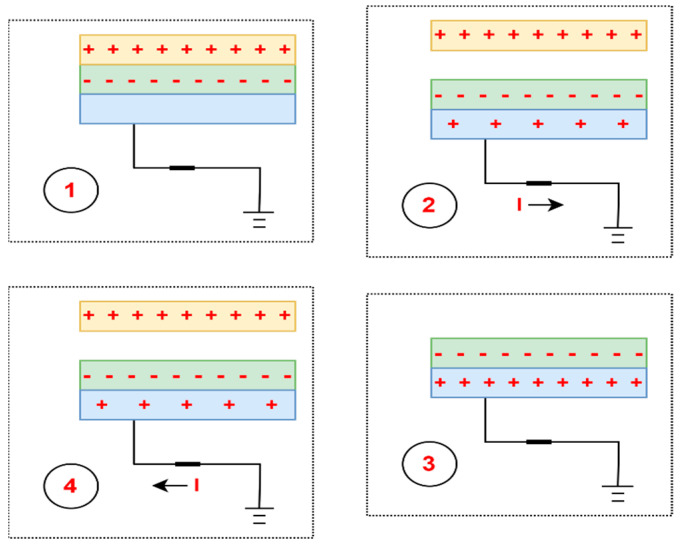
Working mechanism of single-electrode mode. Running order from 1 to 4. I—current.

**Figure 6 sensors-24-01069-f006:**
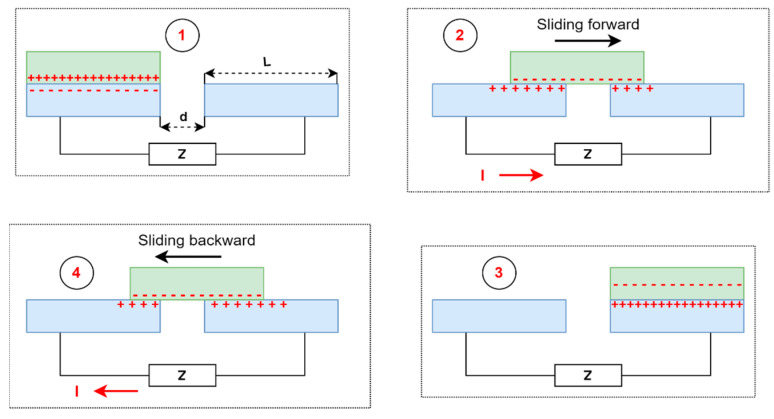
Working mechanism of freestanding triboelectric-layer mode. Running order from 1 to 4. Z—resistor; I—current; L—length of an electrode; d—distance between the electrodes.

**Figure 7 sensors-24-01069-f007:**
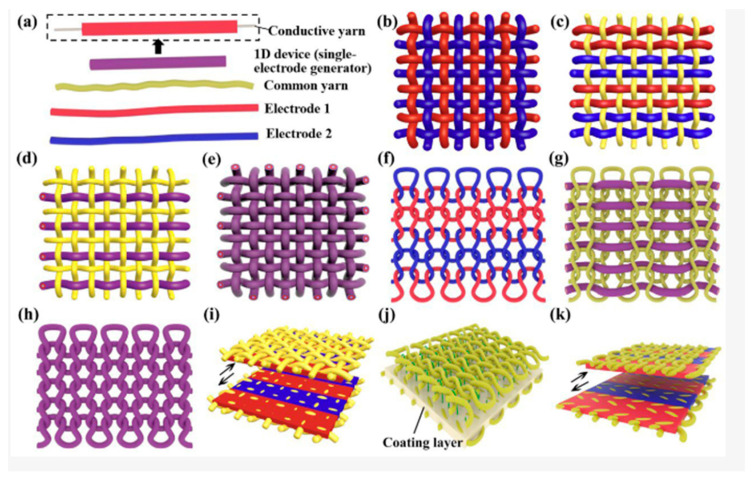
Schematic of the textile-based generators used in human clothing. (**a**) The basic elements of a textile-based generator. (**b**) Woven-structure generator based on two kinds of 1D electrodes. (**c**) Woven-structure generator based on two kinds of 1D electrodes and common yarns. (**d**) Woven-structure generator based on two kinds of 1D devices and common yarns. (**e**) Woven-structure generator based on 1D devices. (**f**) Knitted-structure generator based on two kinds of 1D electrodes. (**g**) One-dimensional device sewn into common knitted fabric. (**h**) Knitted-structure generator based on 1D devices. (**i**) Woven-structure generator based on fabric coated with functional layers. (**j**) Three-dimensional spacer fabric-structure generator and (**k**) knitted-structure generator based on fabric coated with functional layers. Reprinted with permission from [79]. Copyright 2023 American Chemical Society.

**Figure 8 sensors-24-01069-f008:**
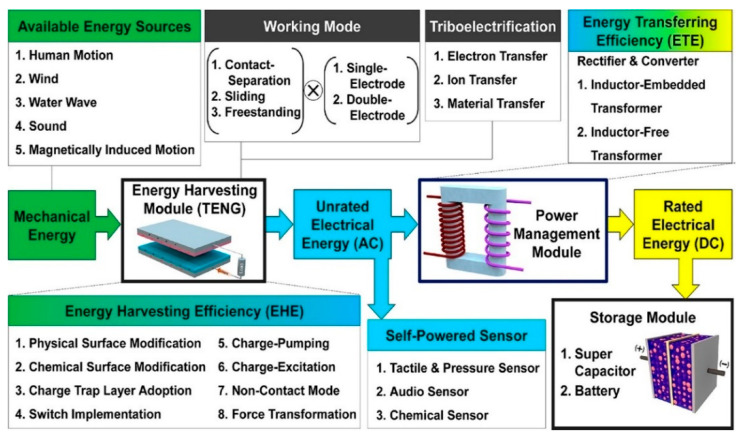
TENG-based energy harvesting system as a block diagram. Reprinted with permission from [127]. Copyright 2023 American Chemical Society.

**Figure 9 sensors-24-01069-f009:**
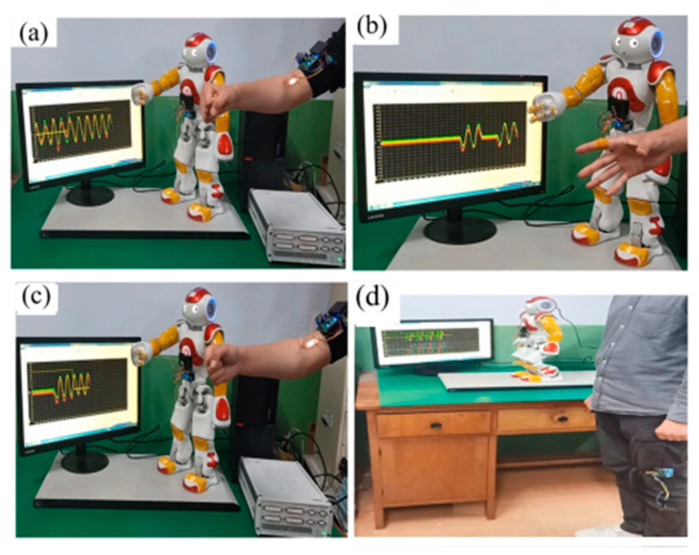
Behaviour-triggering and single-joint HMI tracking test. (**a**) Motion tracking test with/without feedback adjustment. (**b**) Human and robot finger-grab test. (**c**) SPES-assisted detection test when the robot HRPS fails. (**d**) Imbalance detection and fall pre-protection during robot walking. Reprinted with permission from [134]. Copyright 2023 American Chemical Society.

**Figure 10 sensors-24-01069-f010:**
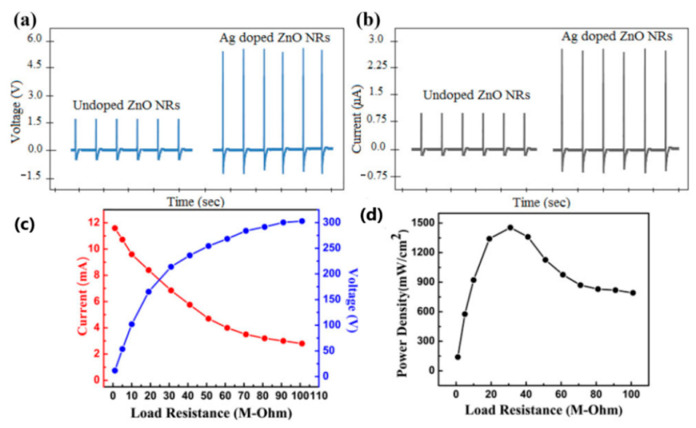
Electrical output performance with (**a**) an output voltage and (**b**) an output current of the undoped ZnO nanorods and Ag-doped ZnO nanorod PENG. (**c**) The behaviour of output current and voltage against RL (Ag-doped ZnO PENG) and (**d**) The output power as a function of RL (Ag-doped ZnO PENG). PENG: piezoelectric nanogenerator; Ag: silver; ZnO: zinc oxide. Reprinted with permission from [138]. Copyright 2023 CC BY 4.0.

**Figure 11 sensors-24-01069-f011:**
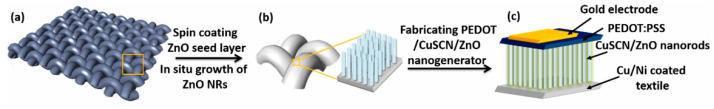
The fabrication process of PEDOT:PSS/CuSCN/ZnO textile nanogenerator. (**a**) Cu/Ni coated textile, (**b**) enlarged textile section with grown ZnO nanorods, (**c**) schematic of nanogenerator. Reprinted with permission from [139]. Copyright 2023 Elsevier Ltd.

**Figure 12 sensors-24-01069-f012:**
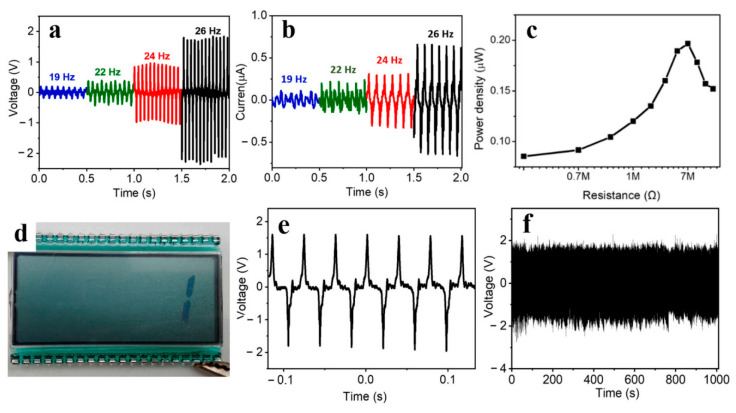
(**a**) Voltage-time and (**b**) current-time scan of PEDOT:PSS/CuSCN/ZnO textile nanogenerator with 3-ZnO nanorods, by shaking at 19 Hz, 22 Hz, 24 Hz and 26 Hz, (**c**) the corresponding power density test by shaking at 26 Hz, with the peak output of 0.2 µW at 7 MΩ corresponding to 0.38 µW/cm^2^, (**d**) LCD display powered by the nanogenerator under shaking at 26 Hz, displaying number “1” on the screen; stability test of the PEDOT:PSS/ZnO/CuSCN textile nanogenerator after six months (**e**) and durability during 1000 s test with over 20,000 cycles (**f**) by shaking at 26 Hz, confirming high reliability of fabricated nanogenerators. Reprinted with permission from [139]. Copyright 2023 Elsevier Ltd.

**Figure 13 sensors-24-01069-f013:**
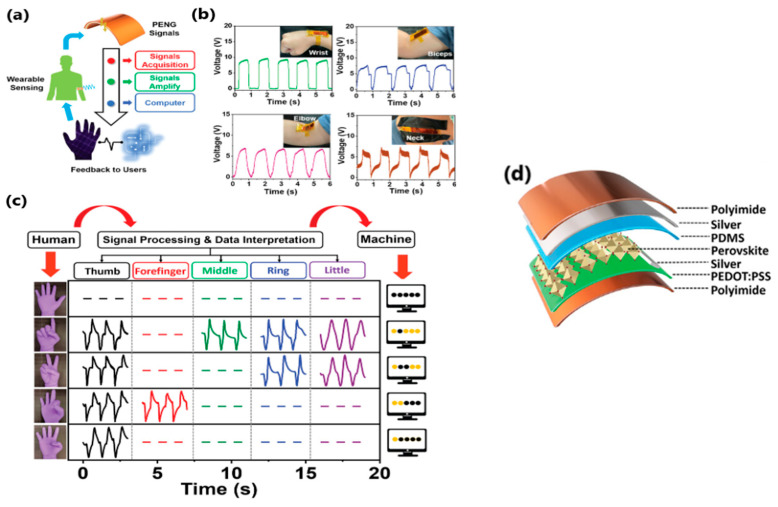
Demonstration of a human–machine-interface application by MN-PENGs. (**a**) Schematic illustration of the human–machine-interface operation. (**b**) Voltage signals from wrist, biceps, elbow, and neck. (**c**) Signals in the smart gesture-recognition system. (**d**) Schematic illustration of the configuration of MN-PENG. Reprinted with permission from [143]. Copyright 2023 Wiley-VCH.

**Figure 14 sensors-24-01069-f014:**
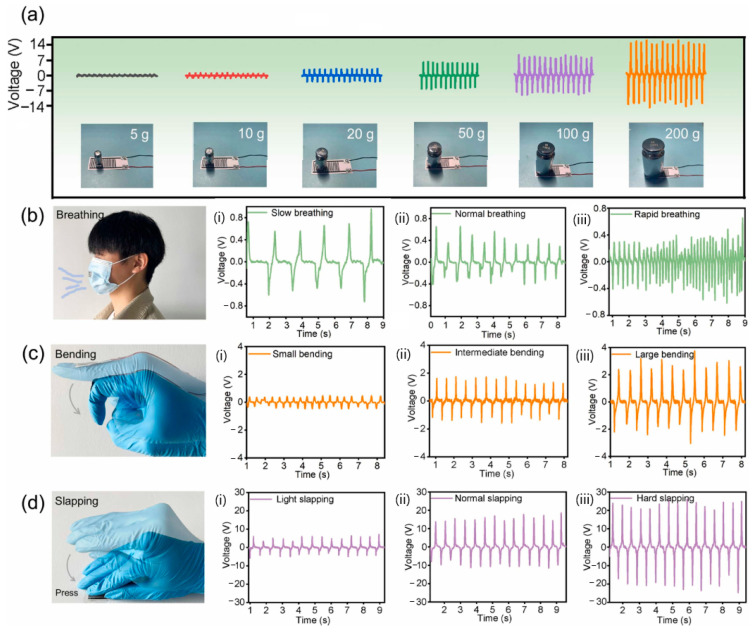
The characteristic output signal of the sensor: (**a**) under different weight pressure; (**b**) breathing at different frequencies; (**c**) bending fingers at different angles; (**d**) slapping at different strengths. Reprinted with permission from [144]. Copyright 2023 Elsevier Ltd.

**Figure 15 sensors-24-01069-f015:**
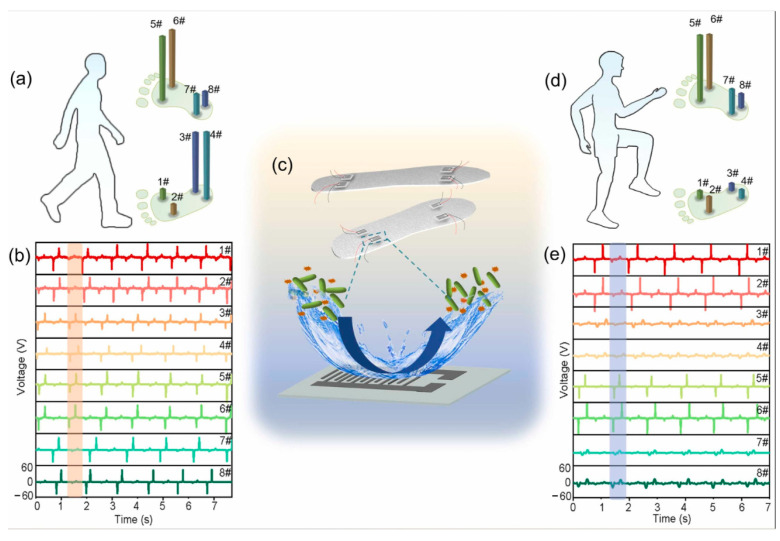
Human gait monitoring. (**a**) Schematic diagram of the human body when walking straight, and the corresponding pressure distribution on the four sensors distributed on the insoles; (**b**) Output signals of the sensors at different locations during straight walking; (**c**) Schematic diagram of the integration of superhydrophobic and anti-fouling sensors on the insole; (**d**) Schematic diagram of the human body when walking in place, and the corresponding pressure distribution on the four sensors distributed on the insole; (**e**) Output signals of the sensors at different locations during in-place walking. Placement on the left foot—1#, 2# − ball of the foot, 3#, 4# − on the heel; placement on the right foot—5#, 6# − ball of the foot, 7#, 8# − on the heel. Reprinted with permission from [144]. Copyright 2023 Elsevier Ltd.

**Figure 16 sensors-24-01069-f016:**
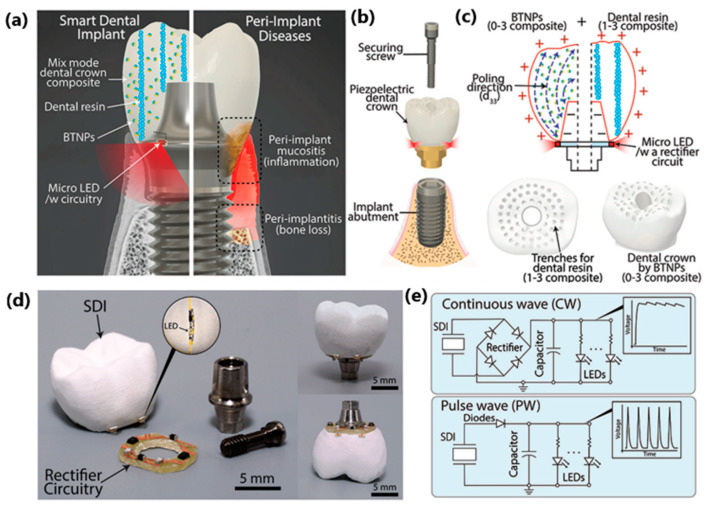
Smart Dental Implant System. (**a**) Ambulatory photo-biomodulation therapy enabled by SDI maintains overall oral health, while normal dental implants without therapeutic function can cause severe oral diseases (image was modified from the source: http://www.deardoctor.com/articles/peri-implantitis-can-cause-implant-failure/). (**b**) Schematic view of SDI assembly based on a screw-retain dental implant design, consisting of (**c**) two-phase composite dental crown, associated electronics, and micro-LEDs. (**d**) Prototype SDI on a US penny. (**e**) Two different types of integrated circuits for continuous wave (CW) or pulsed wave (PW). Reprinted with permission from [148]. Copyright 2023 Wiley-VCH.

**Figure 17 sensors-24-01069-f017:**
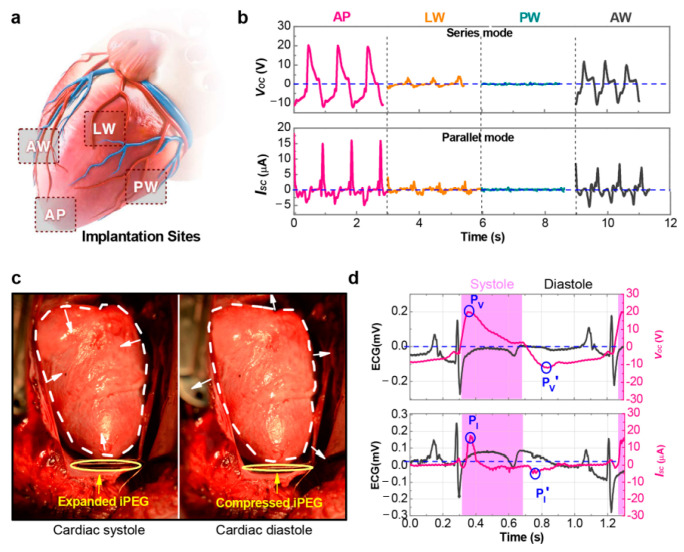
In vivo evaluation for the performance of the iPEG. (**a**) Schematic view illustrating different implantation sites: apex (AP), anterior wall (AW), posterior wall (PW), and lateral wall (LW). (**b**) *V*_oc_ of the implanted iPEG in series mode (upper) and *I*_sc_ in parallel mode (lower) from different implantation sites. The output *V*_oc_ and *I*_sc_ reaches the peak values of 20 V and 15 μA, respectively, when the iPEG is fixed at AP. (**c**) Photographs of the iPEG in the pericardial sac fixed at AP, show that the iPEG expands during the cardiac systolic phase (**left**) and then is compressed by the heart during the cardiac diastolic phase (**right**). (**d**) Magnified and overlapped views of the ECG and the corresponding *V*_oc_ (upper) and *I*_sc_ (lower) waveforms of the iPEG at AP. Reprinted with permission from [151] Copyright 2023 American Chemical Society.

**Figure 18 sensors-24-01069-f018:**
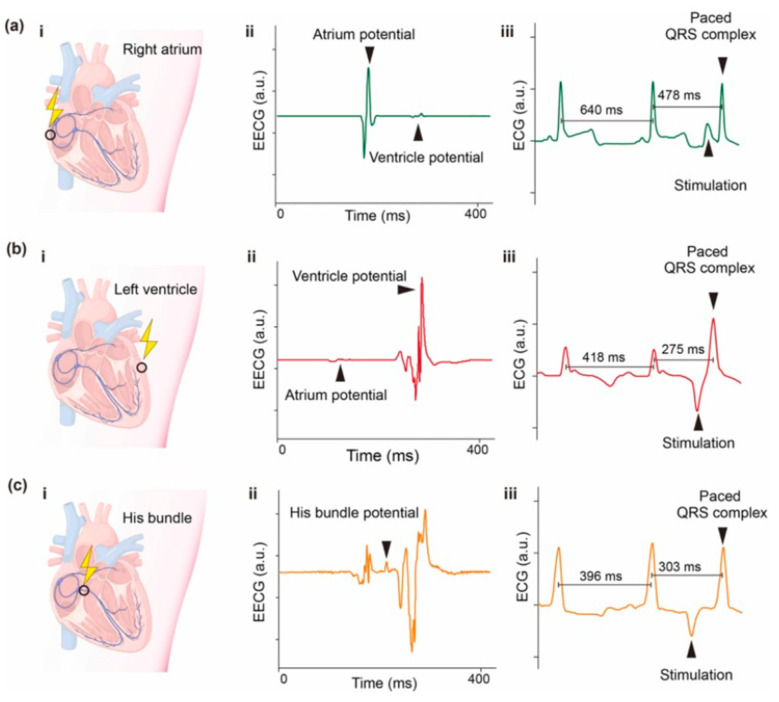
Self-powered cardiac pacing in vivo. (**i**) Schematic diagram, (**ii**) epicardium electrocardiogram (EECG) of pacing sites, and (**iii**) typical valid pacing ECG at (**a**) right atrium, (**b**) left ventricle, and (**c**) His bundle. Reprinted with permission from [152]. Copyright 2023 Elsevier Ltd.

**Figure 19 sensors-24-01069-f019:**
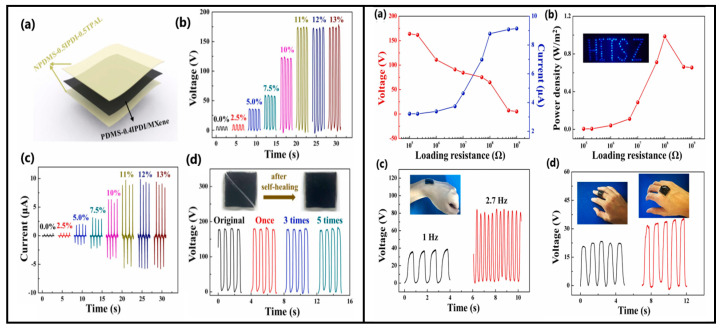
(**Left**) Assembly and output performance of TENGs. (**a**) Sandwich-like structure. (**b**,**c**) Output voltage and current vs. MXene doping content. (**d**) Output voltages of DCS-TENG for different self-healing cycles. Reprinted with permission from [157]. Copyright 2023 Elsevier Ltd. (RIGHT) Application of DCS-TENG. (**Right**) (**a**) Output voltages and currents varied with applied resistances. (**b**) Power densities varied with applied resistances. The inset is 80 LEDs lit by DCS-TENG. (**c**) Output signals when the wrist is rotated slowly and rapidly, respectively. (**d**) Output signals of DCS-TENG are attached to different body positions. Reprinted with permission from [156]. Copyright 2023 Elsevier Ltd.

**Figure 20 sensors-24-01069-f020:**
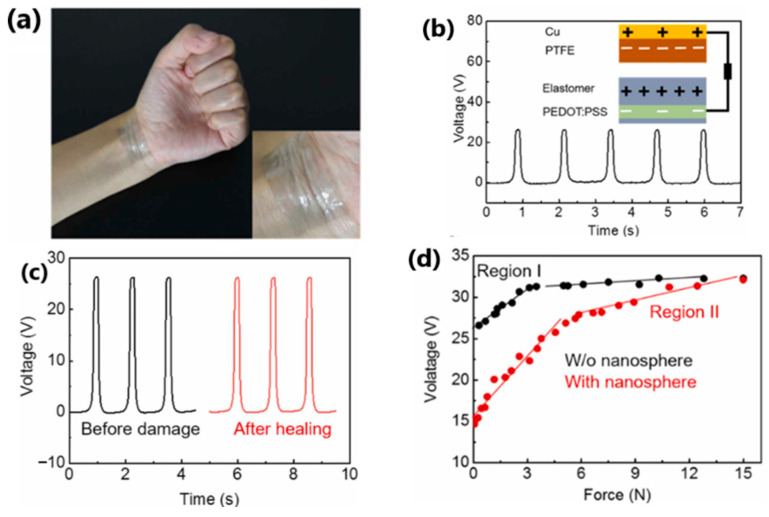
Design, optical and electrical characteristics of the ultrathin, highly transparent, and self-healing triboelectric nanogenerators (TENGs). (**a**) Photograph of the triboelectric skin attached to a wrist. Inset is a magnified image of the device perfectly fitted onto the wrinkles of the wrist. (**b**) Voltage of the TENG working in a contact–separation mode with PTFE/Cu, driven by a linear motor (area: 4 cm^2^; frequency: 1 Hz; force: 5 N). Inset is the TENG’s schematic diagram and working principle, where self-healing elastomer is used as the triboelectric layer and PEDOT:PSS is used as the current collector. (**c**) The output voltages of the TENG before mechanical damage and after healing. (**d**) The influence of the mechanical force on the output voltages of the TENGs with and without nanosphere. Reprinted with permission from [157]. Copyright 2023 Elsevier Ltd.

**Figure 21 sensors-24-01069-f021:**
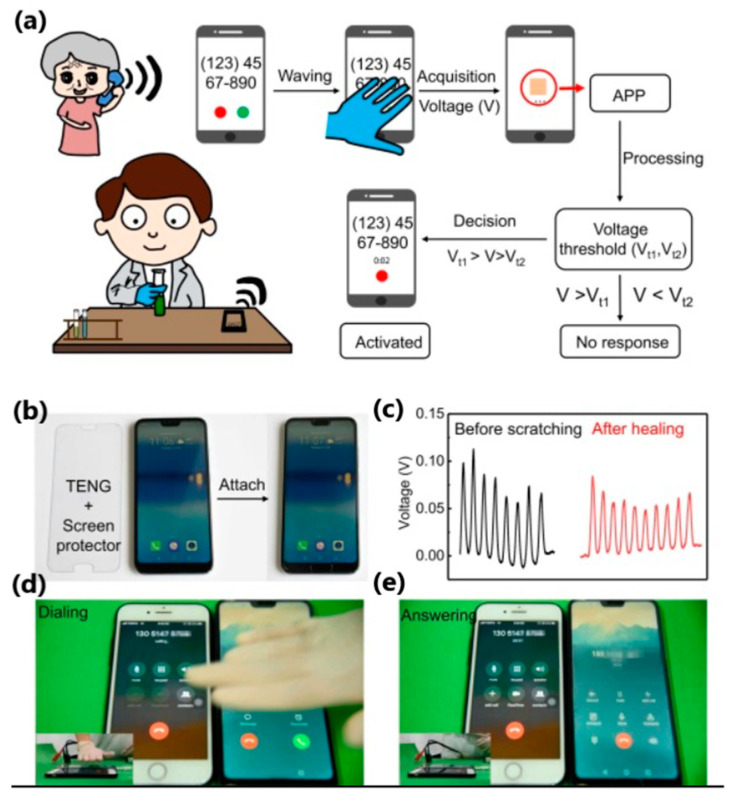
Non-contact gesture-control application. (**a**) Schematic drawing and process flow of the non-touch gesture-control system. (**b**) Photographs of the ultrathin TENG on a screen protector before and after being attached to a mobile phone. (**c**) Induced voltage signal from the screen before scratch and after healing, by waving the hand with the glove above the screen at a distance of ~2 cm. (**d**) Waving the hand above the screen during a phone call. (**e**) Answering the phone call after the waving. Reprinted with permission from [157]. Copyright 2023 Elsevier Ltd.

**Figure 22 sensors-24-01069-f022:**
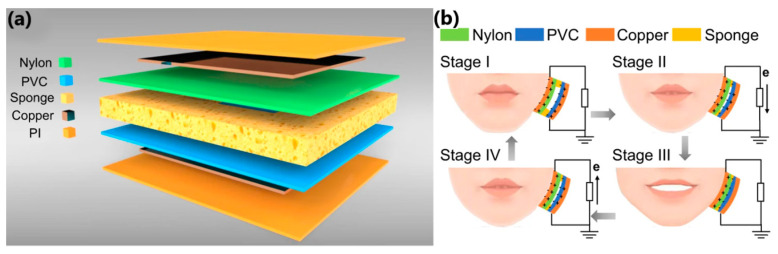
(**a**) Structure scheme for the flexible triboelectric sensor. (**b**) Schematic diagram of four stages of charge transfer in one mouth open–close cycle. The mouth-opening process pushes the sensor, and the mouth-closing process releases the sensor, resulting in the current flow in opposite directions. Reprinted with permission from [158]. Copyright 2023 Springer Nature.

**Figure 23 sensors-24-01069-f023:**
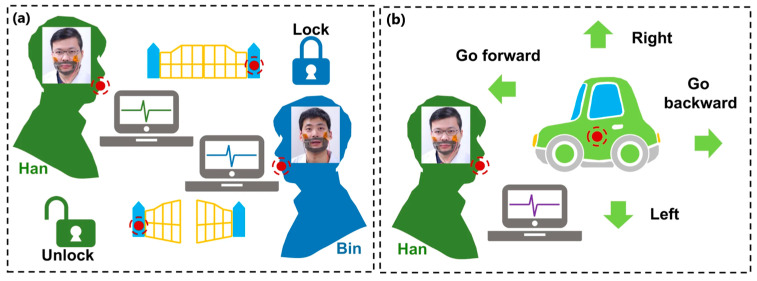
The applications for lip-language decoding in personal identity verification (PIV), toy-car control and lip motion-to-speech conversion assist with communication for people lacking a voice. (**a**) Schematic diagram of unlocking a gate by lip motion with personal identity verification. (**b**) Schematic diagram of direction control for toy car motion by lip motion. Reprinted with permission from [158]. Copyright 2023 Springer Nature.

**Figure 24 sensors-24-01069-f024:**
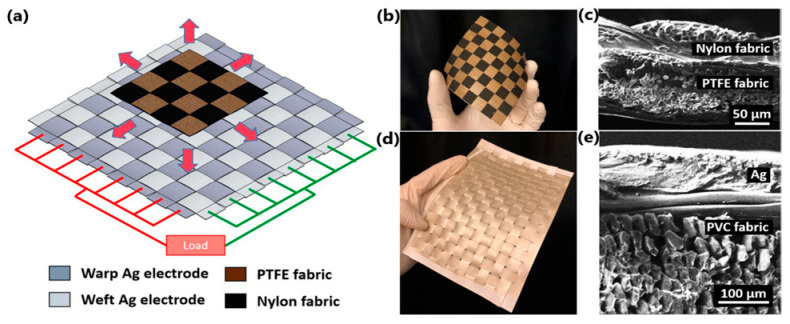
Schematic illustration of (**a**) woven-TENG for *N* = 4 with nylon fabric as positive material, PTFE fabric as negative material and woven Ag electrodes. (**b**) Photograph and (**c**) SEM image of top substrate of the woven-TENG comprising the PTFE and nylon fabric for *N* = 8. (**d**) Photograph and (**e**) SEM image of the bottom substrate of the woven-TENG comprising the Ag-coated PVC fabric as the electrodes. Reprinted with permission from [159]. Copyright 2023 Elsevier Ltd.

**Figure 25 sensors-24-01069-f025:**
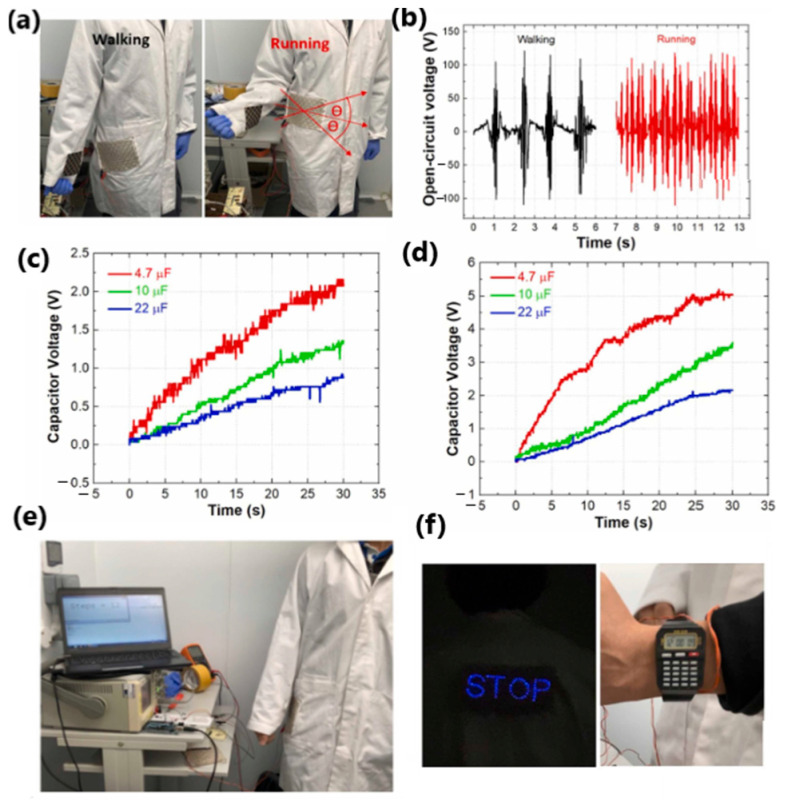
(**a**) Photographs of woven-TENG embedded into a lab coat for harvesting energy from running and walking. (**b**) Transient VOC during running and walking. Transient VC for different capacitors charged during (**c**) running and (**d**) walking. (**e**) Photograph presenting the use of the woven-TENG as a sensor for step counting via arm motion (pedometer). (**f**) Wearable night-time warning indicator for pedestrians and digital watch powered by woven-TENG output during running. Reprinted with permission from [159]. Copyright 2023 Elsevier Ltd.

**Figure 26 sensors-24-01069-f026:**
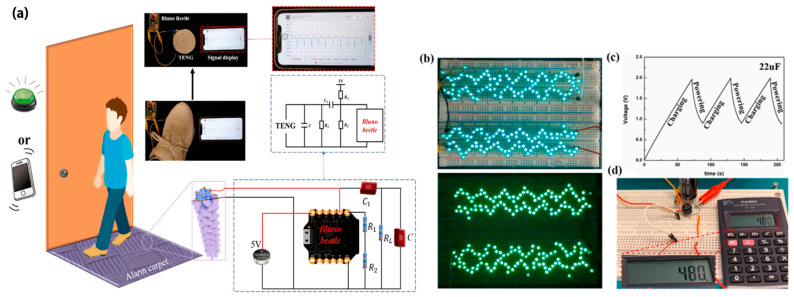
(**a**) Schematic diagram of anti-theft alarm carpet. (**b**) Lighting up 200 LEDs under different light environments. (**c**) A commercial capacitor’s charging/discharging curves connected with CSCY-TENG to power a commercial calculator intermittently. (**d**) The photographs of a commercial electrical calculator powered by CSCY-TENG. Reprinted with permission from [160]. Copyright 2023 Elsevier Ltd.

**Figure 27 sensors-24-01069-f027:**
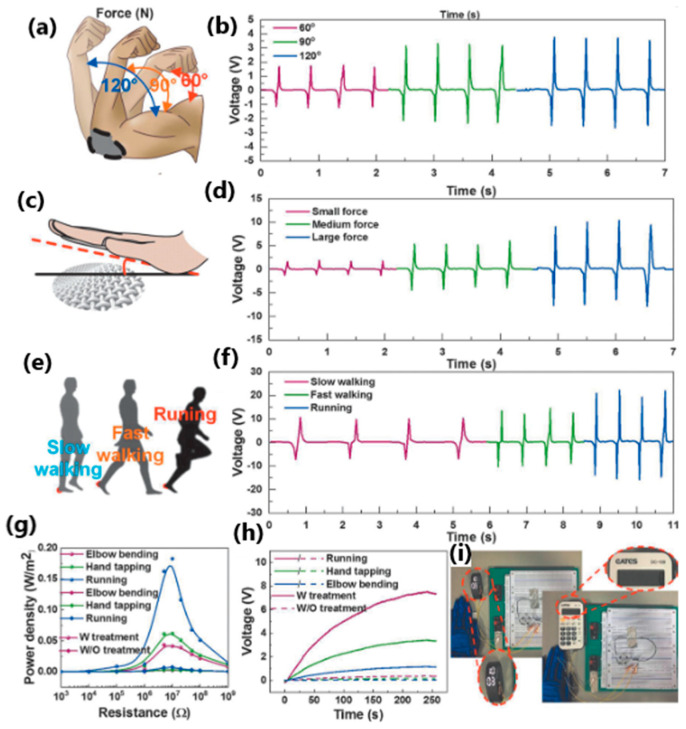
The biomechanical energy harvesting of the superhydrophobic textile TENG. (**a**) The schematic diagram of the device attached on the human elbow to harvest the elbow-bending energy. (**b**) The output voltage with bending angle of 60°, 90°, 120°. (**c**) The schematic diagram of energy harvesting based on hand tapping. (**d**) The output voltage with small, medium and large force. (**e**) The schematic diagram of energy harvesting based on walking and running. (**f**) The output voltage with slow walking, fast walking and running. (**g**) The power curves of elbow bending, hand tapping, and running by using treated textile and untreated textile. (**h**) The charging curves of elbow bending, hand tapping, and running. (**i**) The photographs of powering an electronic watch and calculator using the stored electrical energy in a 10 µF capacitor by biomechanical energy harvesting. Reprinted with permission from [161]. Copyright 2023 Wiley-VCH.

**Figure 28 sensors-24-01069-f028:**
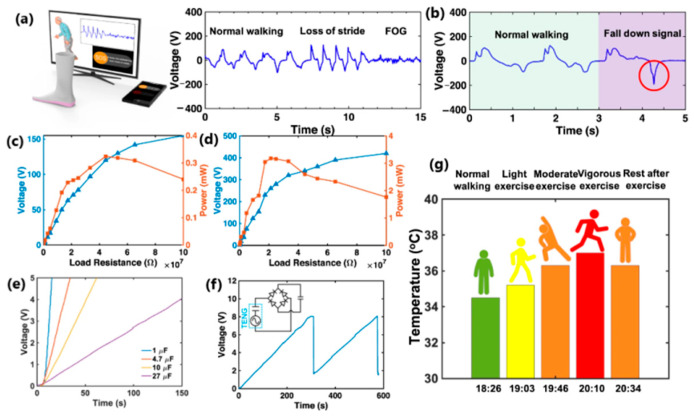
The characterization of T-TENG socks and preliminary gait analysis. (**a**,**b**) The real-time healthcare monitoring of mimetic walking pattern of Parkinson’s disease patient and gait signals of a fall-down event. (**c**,**d**) The maximum output powers of a single sock on the right foot under 1 Hz walking and 2 Hz running were tested by changing the external load resistances from 0.1 to 100 MΩ. (**e**) Charging curve of different capacitors (i.e., 1, 4.7, 10, and 27 μF) were charged to 5 V. (**f**) Charging and discharging curve with the socks on the foot, where each voltage drop represents a discharging to the Bluetooth module. (**g**) Monitoring the temperature under the armpit under various exercise intensities by Bluetooth module. Reprinted with permission from [163]. Copyright 2023 Springer Nature.

**Figure 29 sensors-24-01069-f029:**
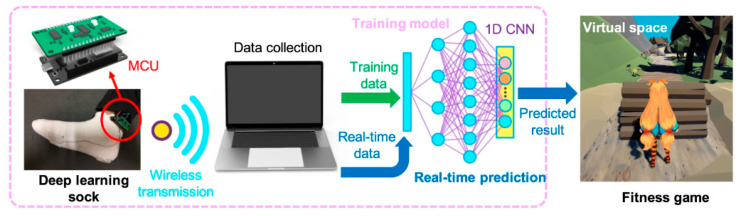
Human-activity recognition of deep-learning-enabled socks. The process flow from sensory information collection to the real-time prediction in a VR fitness game. Reprinted with permission from [163]. Copyright 2023 Springer Nature.

**Figure 30 sensors-24-01069-f030:**
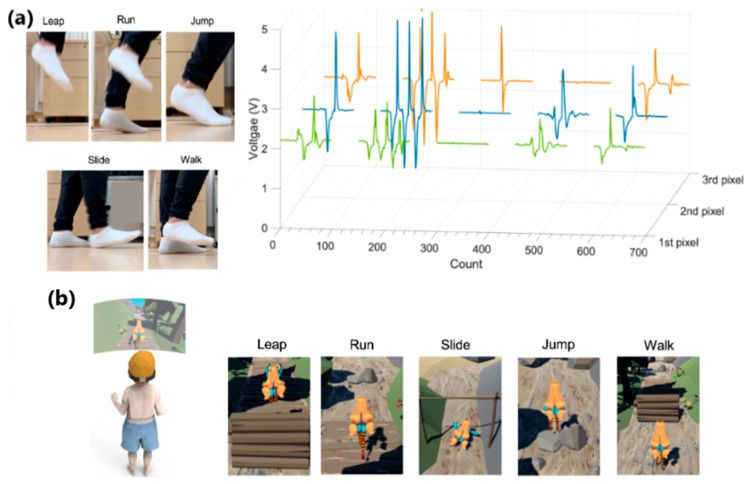
Human-activity recognition of deep-learning-enabled socks. (**a**) The 3D plots of the deep-learning-sock outputs responding to different motions (leaping, running, sliding, jumping, and walking). (**b**) The motion of the virtual character corresponding to real motion in a proposed digital human system. Reprinted with permission from [163]. Copyright 2023 Springer Nature.

**Figure 31 sensors-24-01069-f031:**
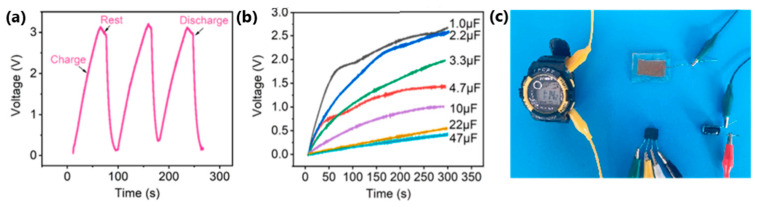
Demonstration of the E-TENG for energy harvesting. (**a**) Voltage real-time charge/discharge profile of a 22 μF capacitor connected to power management. (**b**) Charging-capability curves of the E-TENG under different capacitors (1–47 µF). (**c**) A photograph of the E-TENG-based self-charging system for driving an electronic watch. Reprinted with permission from [118]. Copyright 2023 Elsevier Ltd.

**Figure 32 sensors-24-01069-f032:**
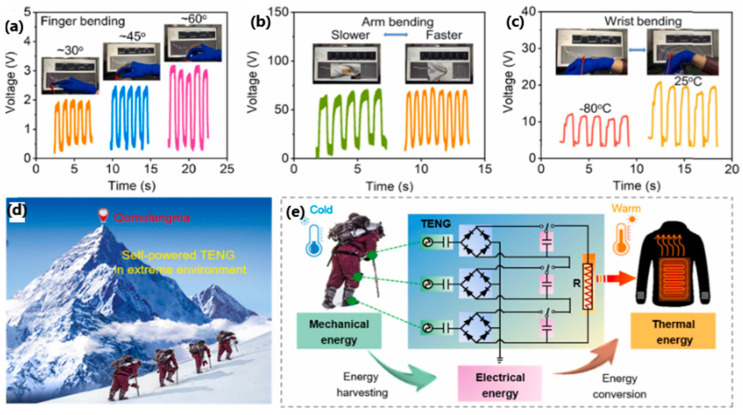
Applications of E-TENG-based energy harvester for green mechanical energy and self-powered sensor to monitor body movements. Photographs and open-circuit voltage signals of the E-TENG attached on body parts to monitor (**a**) finger bending under different bending angles, (**b**) arm bending under different bending frequencies and (**c**) wrist bending under different temperatures (−80 and 25 °C). (**d**) Generated electrical energy of the E-TENG for wearable applications in extreme environments. (**e**) Detailed pictures and the equivalent circuit of the self-charging system for practical applications. Reprinted with permission from [118]. Copyright 2023 Elsevier Ltd.

**Figure 33 sensors-24-01069-f033:**
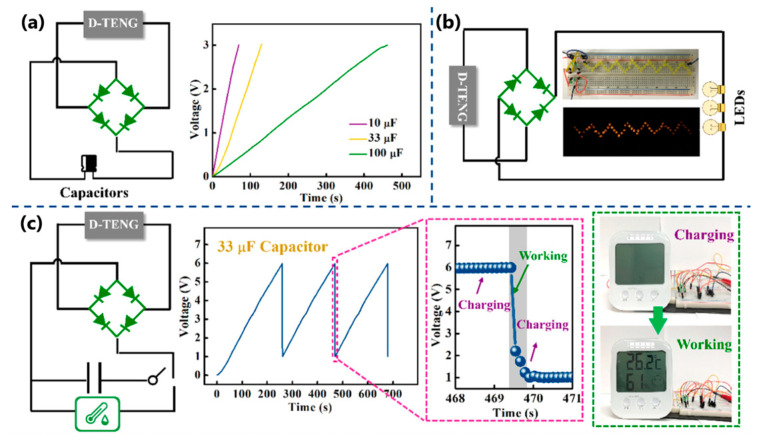
(**a**) Equivalent circuit of capacitor charging and charging behaviour of the D-TENG with 10, 33, and 100 μF capacitors. (**b**) Equivalent circuit and photographs of 52 yellow lit LEDs. (**c**) Equivalent circuit of powering a hygrometer thermometer and the voltage profile of 10 μF capacitor being charged by the D-TENG and used to drive a hygrometer thermometer, and photographs of the hygrometer thermometer in a working state. Reprinted with permission from [119]. Copyright 2023 American Chemical Society.

**Figure 34 sensors-24-01069-f034:**
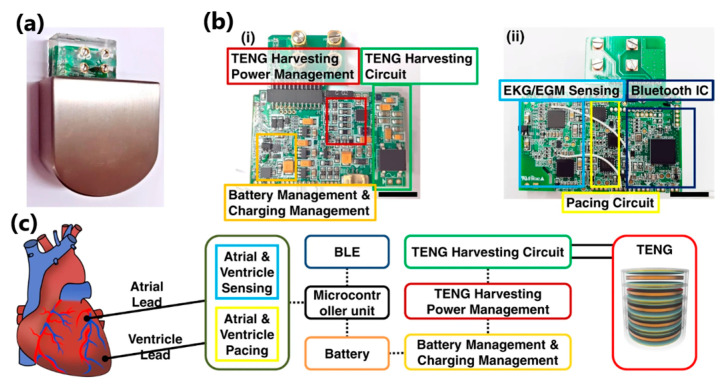
Self-rechargeable cardiac pacemaker system. (**a**) Optical image of the self-rechargeable cardiac pacemaker. (**b**) Photographs of the (**i**) front and (**ii**) back of the integrated system. Scale bar, 1 cm. (**c**) Functional block diagram of the self-rechargeable cardiac pacemaker. Reprinted with permission from [165]. Copyright 2023 Springer Nature.

**Figure 35 sensors-24-01069-f035:**
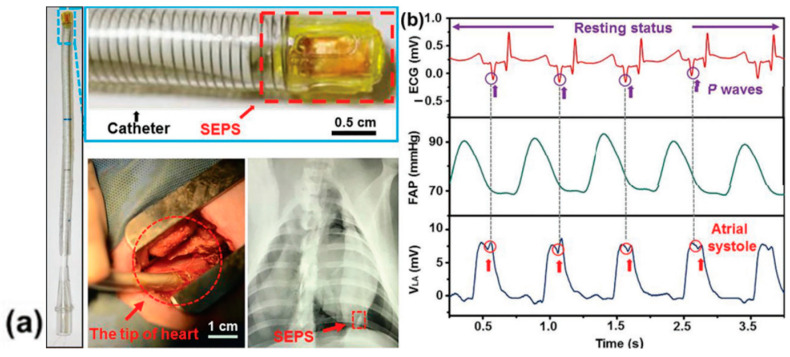
(**a**) Photograph of minimally invasive surgery with a DR image of the heart implanted with a device by integration with a surgical delivery system. (**b**) Detailed inspection into the corresponding relationship between waveforms of ECG and the SEPS outputs. (**c**) The comparison among signals of ECG, FAP, and SEPS during the reinforcing process of cardiac function. (**d**) Ectopic R waves in representative ECG indicating ventricular premature contraction corresponded to an enhanced waveform of the device. (**e**) Disorganized waveforms of SEPS signals with quickened frequency were observed when ventricular fibrillation occurred. Reprinted with permission from [166]. Copyright 2023 Wiley-VCH.

**Figure 36 sensors-24-01069-f036:**
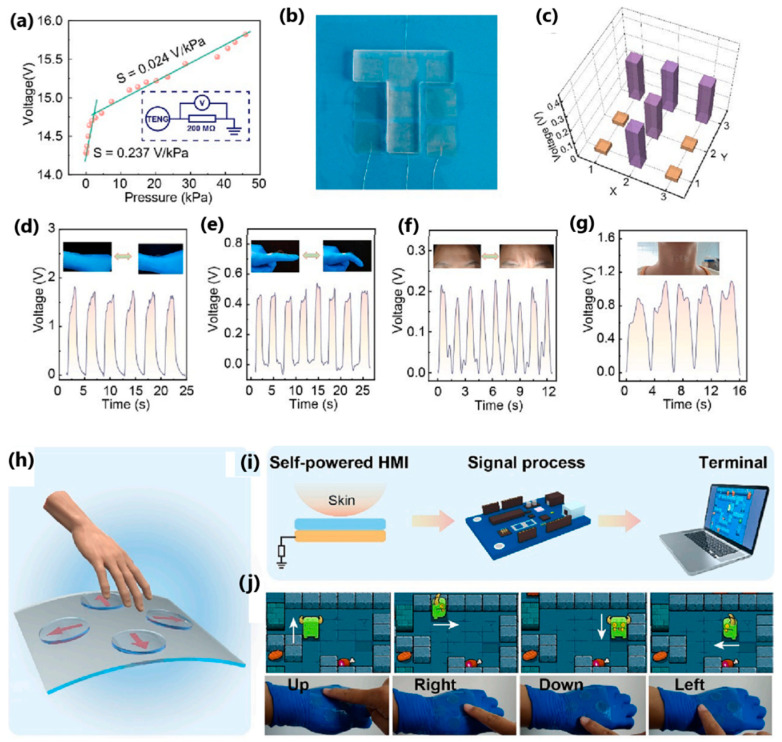
Application for self-powered tactile sensing and human–machine interaction. (**a**) Output voltage of the sensor under different applied pressures. The inset diagram displays the TENG-based tactile sensor scheme with a resistor of 200 MΩ. (**b**,**c**) Photograph and corresponding 3D voltage distribution of the sensor-array mapping pressed with a T-shaped acrylic plate. (**d**–**g**) Real-time voltage signals measured by the tactile sensor for the movement of (**d**) wrist, (**e**) finger, (**f**) forehead, and (**g**) throat. (**h**) Schematic diagram of a flexible transparent self-powered HMI system for game control. (**i**) Working diagram of an HMI system. (**j**) Demo of “Little Monster”. The four sensing units can detect pressure and control the movement of the “little monster”. Reprinted with permission from [169]. Copyright 2023 Elsevier Ltd.

**Figure 37 sensors-24-01069-f037:**
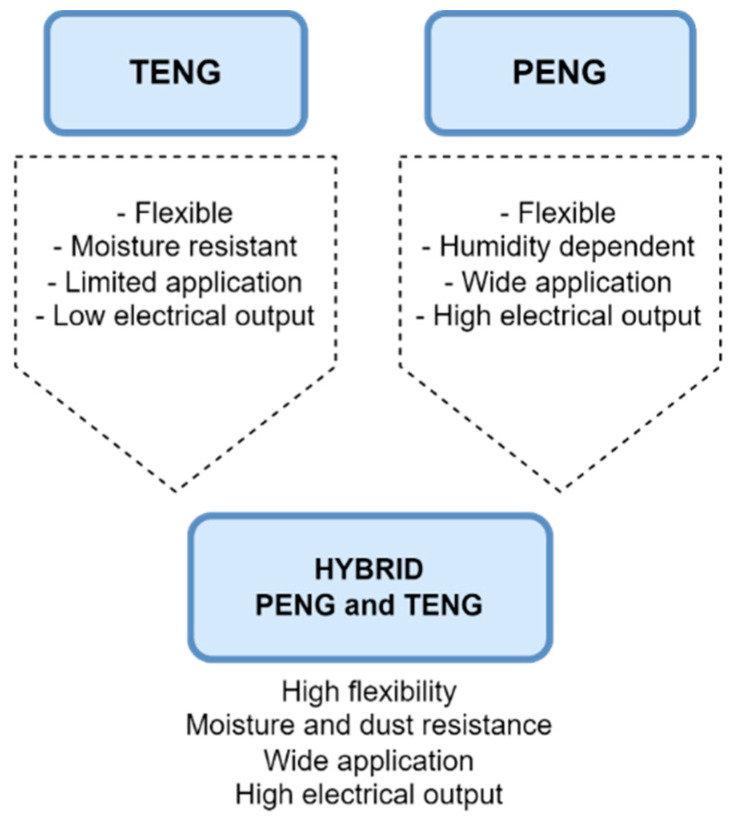
Advantages of using the hybrid PENG-TENG system [153].

**Figure 38 sensors-24-01069-f038:**
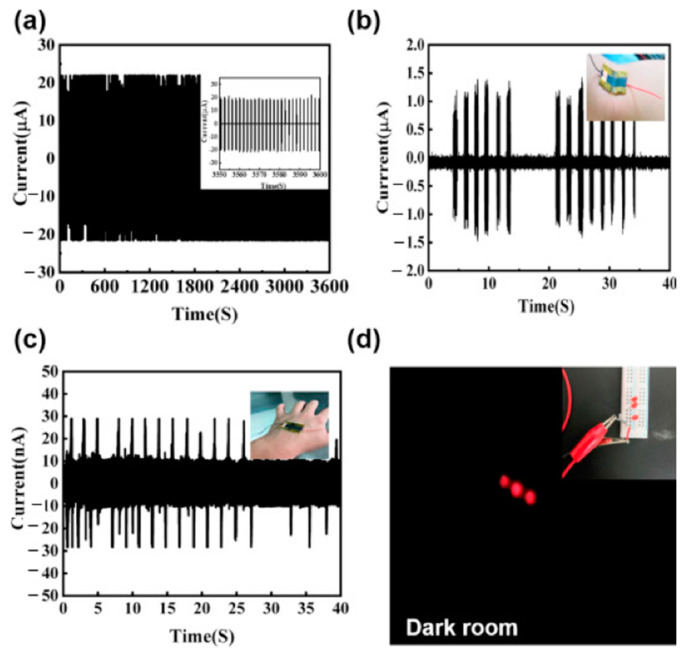
(**a**) Durability test results conducted to confirm the mechanical stability of the H–P/TENG. (**b**,**c**) Output current of the H–P/TENG for real-time posture monitoring and harvesting body-motion energy. Insets shows the corresponding photography images for the tests. (**d**) Photograph showing three LEDs lit up by the electricity generated from the fabricated H–P/TENG. Reprinted with permission from [172] Copyright 2023 Elsevier Ltd.

**Table 1 sensors-24-01069-t001:** Energy harvesting techniques.

Source	Human Body Energy	Energy Harvesting Technique
Chemical Energy	Glucose Lactate	Biofuel Cell
Thermal Energy	Body Temperature Evaporation Heat	Thermoelectric Generator
	Evaporation Heat Respiration Heat	Pyroelectric Generator
Biomechanical Energy	Body Motion	Piezoelectric Generator
	Heart Beat	Triboelectric Generator
	Respiratory Movement	Electromagnetic Generator

**Table 2 sensors-24-01069-t002:** Selected advantages and disadvantages of each piezoelectric transducer.

Configuration Type	Advantages and Characteristics	Disadvantages
Unimorph or bimorph cantilever beam	Simple structure Low cost of fabrication Low resonance frequency High mechanical performance	Cannot resist a high impact force
Cymbal transducer	High energy output Can withstand high impact force	Limited applications
Circular diaphragm	Compatible with pressure working mode	Stiff High resonance frequency
Stacked configuration	Can withstand high mechanical load Compatible with pressure working mode	High stiffness

**Table 3 sensors-24-01069-t003:** Summary of the most important properties for piezoelectric and triboelectric nanogenerators [16].

		Piezoelectric Nanogenerator	Triboelectric Nanogenerator
Materials [175]		BaTiO_3_ and P(VDF-TrFE), AlN, polymer threads, ZnO nanowires and nanorods, PZT nanowires and ceramic, PVDF, PVDF-NaNbO_3_, PVDF polymer and nanofibers, Ceramic PMNZT, PTFE and Al, Al wires and PDMS	PTFE, FEP, PET, PDMS Metals—Cu, Al, Ni, Graphene Nitrile, silicone, Kapton film, PLGA-PCL, Silk fibroin, cellulose, chitin, Rice paper, egg white
Structure [176]		Spiral twining/multilayer/winding Coaxial Core-sheath Woven (2D and 3D) Twisted Sandwich E-skin	Nanofiber stacking/embedded Coaxial Core-shell Knitted/woven (2D and 3D) Textile, fibre, yarn 3D printed Sandwich E-skin
Output performance	Pulse sensor	97.5 V; 1.16 µA [177]	109 V; 2.73 µA [19]
	E-skin	3.2 V; 56.1 nA [133]	220 V; 1.12 µA [178]
	Cardiac pacemaker	3.5 mV; 60 nA [179]	136 V; 2 µA [165]
Material fabrication process		− Complex and expensive fabrication process − Ceramics are fragile and brittle	+ Wide selection of materials + Relatively low cost + Easily fabricated
Power density		+ High power density − Lower current density − Affected by operation frequency	+ High power density + Higher output voltage + Multiple-operation mode
Sensing performance		+ High dynamic sensing + Fast response − Poor static sensing	+ Wide sensing range + Fast response − Poor static sensing
Scalability		+ Successfully miniaturized − Expensive and fragile	+ Large-area application − Low-output voltage after miniaturization
Durability		+ Good mechanical durability	+ Good stability of electrical signal − Poor mechanical performance − High erosion of the material − Sensitive to environment
Biocompatibility		+ (Co)polymers and lead-free materials are biocompatible − Ceramics can be toxic	+ Biocompatible

**Table 4 sensors-24-01069-t004:** Summary of PENG, TENG and hybrid PENG-TENG kinetic-energy harvesters in recent years (2019–2023).

Energy Harvester	Materials	Position	Max Output Current (Isc)	Max Output Voltage (Voc)	Max Power Density or Output Power	Function	Ref.
PENG	PE braided layer, PET spacer layer, PTFE braided layer	Hand tapping, shoe insoles, fixed carpets	0.25 μA	32.0 V	2.6 mW/m^2^	Charging capacitors (0.22–4.7 μF), lightning 16 LEDs, walking- and-sitting state monitoring	[180]
	PVDE-TrFE, gold, PI film	Eye, wrist, finger, abdominal area	1.07 nA	9.39 V	2.84 μW/cm^3^	Charging capacitors (4.7, 6.8, 10, 22 nF and 0.1 μF), lightning LED	[181]
	PANI coated pure P(VDF-TrFE) nanofibers, PANI coated P(VDF-TrFE)/BT nanocomposite nanofibers	Arm, knee, wrist, finger, and shoe sole	3 μA	68.0 V	225 mW/m^2^	Charging capacitors (1.0, 4.7, and 10 μF), powering more than 10 LEDs, self-powered wearable sensor	[182]
	PDMS, PZT/epoxy, Ag electrode, Polycarbonate	Sandwiched spirally coiled structure—walking	196 μA	36 V	3.72 mW/cm^2^	Powering 27 LEDs, energy storage (1 and 47 μF), powering hygrometer	[183]
	BaTiO_3_ nanoparticles, P9VDF-TrFE) matrix, silver flake	Mounted on a sock	2 μA	6 V	1.4 μW/cm^2^	Self-powered gait sensor	[184]
	PVDF nanofibers, ZnO flakes and rGO sheets	Implanted in the heart of an adult dog	3 μA	5 V	138 μW/cm^3^	Charging a 100 μF capacitor—powering a battery-free pacemaker	[185]
	PDMS-Ecoflex, LIG/PI, PVDF/CB, PVDF, PVDF/CA, PVDF-PDMS-CA MSc, Parylene C	Face mask—smiling, opening mouth, breathing in/out	0.28 μA	3 V	0.85 × 10^−2^ μW/cm^2^	Smart mask coupled with energy harvesting devices—anti-infection protection	[186]
	PI, ITO, Cr/Al, PET, PDMS + MASnBr_3_ composite	Finger (bending, tapping), heel (tapping, pressing)	13.76 μA/cm^2^	18.8 V	74.52 μW/cm^2^	Charging capacitors (0.11, 2.2, and 4.7 μF)—driving a stopwatch and commercial LED	[187]
TENG	Copper-nickel fabric, PDMS doped with BaTiO_3_ nanoparticles	Shoe insole, energy carpet	137 μA	480 V	486 μW/cm^2^	Shoe insole—lightning 700 LEDs; powering a digital watch, self-powered sensing system, human-falling detection	[188]
	TENG—PDMS, PPy, rGO, PLA; SC—PDMS, MWCNTs, PVA/H_3_PO_4_, PPy, MnO_2_, rGO	Woven into the fabric (forearm)	0.4 μA	50 V	17.9 μW/cm	Lighting 52 LEDs, powering the electronic watch, charging capacitors	[189]
	CMC/WPU/PEI/SiO_2_@TiO_2_ NPs	Hand, sleeve, knee and elbow joints, shoe insole	6 μA	204 V	1.62 W/m^2^	Charging capacitors (2.2, 4.7, 10 and 22 μF), powering electronic watch and calculator, self-powered strain/pressure sensor	[190]
	PVA/P(AM- *co* -AA)-Fe^3+^ DN gel	Cheek, forehead, lower limbs, throat, palm, elbow, index finger, knee joints	1.2 μA	238 V	0.27 W/m^2^	Charging capacitors (1, 4.7, 22, 47, 220 μF), powering 28 LEDs, calculator, body/joint motion detection and monitoring	[191]
	PDMS-PTFE/AgNWs-PVA hydrogel TENG	Fixed onto the fingers and soles of feet, wrist	40 μA	450 V	3.07 W/m^2^	Lighting 360 LEDs, self-powered sensing bracelet	[192]
	PVDS-coated CFP, CU-coated CFP	Patch films (working individually and installed on fingers)	9.3 μA	192 V	736.7 mW/m^2^	Charging capacitors (1–47 μF), powering 62 LEDs, calculator, thermohydrometer and electronic watch, HMI-recognition	[193]
	PET, Kapton, rubber band, Ag, PET/Kapton Spacer	TENG wrapped around finger, implemented into a floating backpack	26 μA	1334 V	0.5 W/m^2^	Lighting 360 LEDs, charging capacitors (1, 4.7, 22, 47 and 100 μF), powering electronic watch, stopwatch and calculator	[194]
	Mylar polyester film sheets, PET foam, Ni/Cu nonwoven polyester, PTFE film on Cu electrode	Miura-Ori tube structure	131 μA	1050 V	40 mW	Powering LCD, calculator, wireless temperature sensor, charging 1200 μF capacitor, commercial Li battery, heart rate monitoring	[89]
Hybrid PENG-TENG	Lead-free perovskite/PVDF-HFP, SEBS	Top and bottom side of a shoe pad	25 μA	290 V	-	Powering 100 LEDs, charging capacitors (1–100 μF)	[195]
	PET, Copper, Kapton, PVDF, Aluminium	Add-on fabric patch (knee, elbow), shoe insole	1–2 μA	1–5 V	-	Illuminating LED, charging a 220 μF capacitor	[196]
	PVDF films, Al electrode, Acrylic	Shoe insole	~3 μA	~120 V	127 μW	Lighting LEDs, wireless sensor network	[197]
	BTO-PDMS, PDMS-PTFE, Ag	Kirigami patch—stretching, pressing, twisting	2 μA	255 V	7.5 W/m^2^	Illuminating 118 LEDs, charging capacitors (0.47, 4.7, 10, 47 μF), charging calculator, sensor	[198]
	PEDOT:PSS, PTFE film, Al, Au/Cr, PZT, Ag, Cu	Cotton socks	4.5 μA	196 V	128 μW/cm^2^	Gait analysis, sweat detection, monitoring physiological signals	[199]
	MoS_2_ on Cu foil, ZnO, PVDF	Heel, elbow, machine vibration	4.6 μA	140 V	256 μW/cm^2^	Powering 33 LEDs, calculator and wristwatch, physiological signal monitoring	[200]
	PTFE yarn, BaTiO_3_/PDMS yarn, AgNW/PDMS yarn, metal Cu wire, Cu coil-spring	Stretching, bend-stretching, squeezing, tapping; knee, elbow, wrist	2 μA	400 V	91.6 mW/m^2^	Charging capacitors (10, 22, 33, 47, 100 μF), powering 130 LEDs, self-powered sensing device	[201]

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
