# Peer review of "A Review of Recent Advances in Human-Motion Energy Harvesting Nanogenerators, Self-Powering Smart Sensors and Self-Charging Electronics"

_sensors, 2024, doi:10.3390/s24041069_

Round 1

Reviewer 1 Report

Comments and Suggestions for Authors

The review article "A review of recent advances in human motion energy harvesting nanogenerators, self-powering smart sensors and self-charging electronics" summarized the recent researches about piezoelectric nanogenerators (PENGs), triboelectric nanogenerators (TENGs) and hybrids for energy harvesting applications. The PENGs and TENGs can realize the function of powering micro electronic devices by harnessing the energy of human motion. The authors present various designs of energy harvesting devices from PENGs and TENGs to hybrid systems, with a description of their important properties. The working principles of such PENGs and TENGs is presented in detail. In additional, the applications of the PENGs and TENGs are introduced well. Overall, the reviewer thinks that the manuscript presents recent trends in human motion energy harvesting nanogenerators, self-powering smart sensors and self-charging electronics. At the same time, however, some important contents are missing. Therefore, several issues need to be addressed before publication in Sensors.

1. This manuscript details the applications of PENGs and TENGs in wearable devices, implantable electronic devices and intelligent sensing. However, biofuel cells and pyroelectric nanogenerators are also capable of harvesting energy from human motion. Please discuss the research progress and application of the above two devices in energy harvesting in detail.

2. Authors can summarize the materials, structure, output performance, advantages and disadvantages of PENGs, TENGs, biofuel cells and pyroelectric nanogenerators.

3. Too much space in the manuscript summarizes the existing literature, and the author should put forward more of his own views and opinions.

4. The TENGs and sensors for body motion detection and implantable use requires biocompatibility and high softness without degradation of output performances. To meet such requirements, many approaches on material components and structural designs of the battery have been reported. The design strategies of the TENGs are introduced in the manuscript, but material components are not well described. The author might refer relevant previous works such as Adv. Energy Mater., 2022, 12, 2102991., Nano Energy, 2023, 118, 108932., npj Flex. Electron., 2023, 7, 13., Nano Energy, 2023, 109, 108231.

Comments on the Quality of English Language

1. English needs editorial help. There are many grammar mistakes and improper expression in the manuscript.

Author Response

Thank you for every suggestion you made in your report. Please find our answers in the attached file.

Reviewer 2 Report

Comments and Suggestions for Authors

The article "A review of recent advances in human motion energy harvesting nanogenerators, self-powering smart sensors, and self-charging electronics" looks specifically at piezoelectric nanogenerators (PENGs), triboelectric nanogenerators (TENGs), and devices that combine both technologies to create self-powered systems that can be worn or implanted. The article is generally well-written, despite being somewhat large (60 pages!). The findings and discussion section has a lot of material, but I feel it may be simplified to 10-30 pages at most. The article currently needs substantial revisions in grammar and formatting. The post appears to be more of an exhaustive report or instructional guide than a professionally prepared review article, in my opinion. Unfortunately, I cannot support this piece in its current form. Nonetheless, I urge that the authors condense the information to 30 pages by reducing the discussion and including a complete summary of the existing scholarly opinions on the subject.

The following are some of the minor observations that the manuscript needs editing:

Line 26: sentence ending full stop missing.

Line 37: Please remove ,, from “Global..industrial”.

Line 58: grammar issue “or lack thereof” ?

Line 92-93: Remove “for example”.

Line 98: “Emery” to “energy”?

Line 108: “Transformed electricity” to “transformed into electricity.

Figure 2: Needs sub caption for (a) and (b) and also label missing. Follow same for rest of figures. Lot of them missing sub captions and also labels!

Line 190: Left align or justify the sentence.

Line 385 and Line 394: Left align or justify the sentence or subheading.

Figure 8: figure texts are too small to read.

Line 484: Figure 8 should be Figure 9. Please check and correct this. Also, label (c) in figure is missing.

Comments on the Quality of English Language

The article "A review of recent advances in human motion energy harvesting nanogenerators, self-powering smart sensors, and self-charging electronics" looks specifically at piezoelectric nanogenerators (PENGs), triboelectric nanogenerators (TENGs), and devices that combine both technologies to create self-powered systems that can be worn or implanted. The article is generally well-written, despite being somewhat large (60 pages!). The findings and discussion section has a lot of material, but I feel it may be simplified to 10-30 pages at most. The article currently needs substantial revisions in grammar and formatting. The post appears to be more of an exhaustive report or instructional guide than a professionally prepared review article, in my opinion. Unfortunately, I cannot support this piece in its current form. Nonetheless, I urge that the authors condense the information to 30 pages by reducing the discussion and including a complete summary of the existing scholarly opinions on the subject.

The following are some of the minor observations that the manuscript needs editing:

Line 26: sentence ending full stop missing.

Line 37: Please remove ,, from “Global..industrial”.

Line 58: grammar issue “or lack thereof” ?

Line 92-93: Remove “for example”.

Line 98: “Emery” to “energy”?

Line 108: “Transformed electricity” to “transformed into electricity.

Figure 2: Needs sub caption for (a) and (b) and also label missing. Follow same for rest of figures. Lot of them missing sub captions and also labels!

Line 190: Left align or justify the sentence.

Line 385 and Line 394: Left align or justify the sentence or subheading.

Figure 8: figure texts are too small to read.

Line 484: Figure 8 should be Figure 9. Please check and correct this. Also, label (c) in figure is missing.

Author Response

Thank you for every suggestion you made in your report. Taking your suggestiond into consideration, our review was shortened significantly. All of the minor observations that you have observed in the manuscript were edited and corrected.  More precise summary was made at the end of our review. Please find our answers in the new version of the manusrcipt. 

Round 2

Reviewer 2 Report

Comments and Suggestions for Authors

The article is looking good now.

Comments on the Quality of English Language

I suggest to do few more rounds of spell and grammar checks before the article goes online.

Author Response

Dear Reviewers

Thank you very much for all your comments and suggestions.

We carefully checked the manuscript several times and corrected all grammatical errors.

We also believe that the final, revised version of our manuscript will be acceptable to MDPI Sensors.

Sincerely,

The Authors